# The Herpes Simplex Virus pUL16 and pUL21 Proteins Prevent Capsids from Docking at Nuclear Pore Complexes

Ethan C. M. Thomas[1], Renée L. Finnen[1], Jeffrey D. Mewburn[2], Stephen L. Archer[2], Bruce W. Banfield[1]*

1 Department of Biomedical and Molecular Sciences, Queen's University, Kingston, Ontario, Canada,
2 Department of Medicine, Queen's University, Kingston, Ontario, Canada

* bruce.banfield@queensu.ca

**Data Availability Statement:** Raw data associated with Figs 2D, 3E, 4B, 4C, 4E, 5B, 5C, 6B and 7E have been uploaded to Dryad and are accessible here: https://doi.org/10.5061/dryad.gtht76hsr

## Abstract

After entry into cells, herpes simplex virus (HSV) nucleocapsids dock at nuclear pore complexes (NPCs) through which viral genomes are released into the nucleoplasm where viral gene expression, genome replication, and early steps in virion assembly take place. After their assembly, nucleocapsids are translocated to the cytoplasm for final virion maturation. Nascent cytoplasmic nucleocapsids are prevented from binding to NPCs and delivering their genomes to the nucleus from which they emerged, but how this is accomplished is not understood. Here we report that HSV pUL16 and pUL21 deletion mutants accumulate empty capsids at the cytoplasmic face of NPCs late in infection. Additionally, prior expression of pUL16 and pUL21 prevented incoming nucleocapsids from docking at NPCs, delivering their genomes to the nucleus and initiating viral gene expression. Both pUL16 and pUL21 localized to the nuclear envelope, placing them in an appropriate location to interfere with nucleocapsid/NPC interactions.

## Author summary

Despite the very high prevalence of HSV infections worldwide and many decades of research focused on these important human pathogens, questions related to fundamental aspects of HSV biology remain unanswered. We have provided insight into the mechanism by which HSV averts a short circuit in virion assembly by preventing the attachment of nascent nucleocapsids to NPCs and the delivery of their genomes back into the infected cell nucleus and, in doing so, have provided answers to a long-standing question in herpesvirus biology.

## Introduction

Herpes simplex virus (HSV) 1 and 2 are highly prevalent human pathogens that belong to the alphaherpesvirus subfamily of the *Herpesviridae*. In 2016, it was estimated that half a billion people were infected with HSV-2 and 3.7 billion people were infected with HSV-1 worldwide

**Funding:** This work was supported by the Canadian Institutes of Health Research (CIHR) operating grant 162162, Natural Sciences and Engineering Research Council of Canada Discovery Grant 04249-2018 and Canada Foundation for Innovation (CFI) award 16389 to BWB. SLA was supported by U.S. National Institutes of Health (NIH) grants NIH 1R01HL113003-01A1, NIH 2R01HL071115-06A1, CFI awards 229252 and 33012, CIHR Foundation Grant CIHR FDN 143261, and the William J. Henderson Foundation. ECMT was supported in part by a R. Samuel McLaughlin Fellowship from Queen's University. The funders had no role in study design, data collection and interpretation, or the decision to submit the work for publication.

**Competing interests:** The authors have declared that no competing interests exist.

[1]. Both viruses can cause a variety of diseases that range from trivial to life-threatening. While antiviral drugs are available to treat these infections, drug resistance is a problem. A deeper understanding of the fundamental steps in HSV replication and assembly is expected to enable the development of new therapeutic strategies to combat disease.

HSV virions are complex assemblies comprised of a lipid envelope containing numerous membrane proteins surrounding a 152–154 kilobase pair linear double-stranded DNA genome that is packaged at high pressure within an icosahedral capsid. Between the capsid and envelope lies the tegument, a subvirion compartment that contains roughly 20 virion encoded proteins and an estimated 50 proteins of cellular origin [2]. Upon entering a cell, the HSV capsid and tegument are deposited into the cytoplasm, where most of the tegument dissociates from the capsid [3–7]. Capsids transit from the periphery of the cell to the nucleus along microtubules by recruiting and utilizing dynein motors from the infected cell as well as kinesin motors contained within the tegument of the infecting virion [8–14]. As capsids are too large to enter nuclei through nuclear pore complexes (NPCs), they dock at NPCs [15–17]. Capsid docking to NPCs is facilitated by the capsid-associated proteins pUL36 and pUL25, which interact with nucleoporins RanBP2 (Nup358) and Nup214, respectively [8, 18–25]. The NPC cytoplasmic filaments are comprised of RanBP2, whereas Nup214 is a component of the NPC cytoplasmic ring [26]. During the docking process, it is thought that the capsid portal, through which viral genomes are packaged into capsids during assembly, is oriented with the central pore of the NPC and this leads to conformational changes in the capsid structure that result in genome ejection into the nucleoplasm [8, 15–17, 21, 22, 24, 25]. Once the viral genome is delivered to the nucleoplasm, transcription of viral genes ensues and viral genome synthesis, capsid assembly, and genome packaging into nascent capsids take place.

Genome-containing capsids, called nucleocapsids, transit from the nucleoplasm to the cytoplasm for the final stages of virion maturation. Translocation of the nucleocapsids from the nucleus to the cytoplasm, referred to as nuclear egress, involves the primary envelopment of the nucleocapsids at the inner nuclear membrane and subsequent fusion of their envelopes with the outer nuclear membrane, delivering the nucleocapsids into the cytoplasm adjacent to the cytoplasmic face of the nuclear envelope [27]. Once in the cytoplasm, the nucleocapsids are transported along microtubules towards membranes derived from the trans-Golgi network, or late endosomes, where the nucleocapsids acquire their final envelope in a process called secondary envelopment [28, 29]. After secondary envelopment, the now mature virions are transported within vesicles to the plasma membrane, where vesicle membranes fuse with the plasma membrane releasing the virions into the extracellular space [30].

An outstanding question in herpesvirus biology is why incoming nucleocapsids, during initial infection, can interact and dock with NPCs, while nascent nucleocapsids that have undergone nuclear egress do not dock at NPCs and deliver their genomes to the nucleus from which they emerged. If egressing nucleocapsids delivered their genomes back into the cell nucleus where they were synthesized it would be detrimental to new virion production by removing cytoplasmic nucleocapsids from the virion assembly pathway. During our ongoing analysis of pUL16 and pUL21 mutants, we noted that the HSV tegument proteins pUL16 and pUL21, which are known to form a heterodimeric complex [31, 32], prevented egressing nucleocapsids from binding to NPCs. Cells infected with HSV strains deleted for pUL16 or pUL21 accumulated capsids at the cytoplasmic face of the nucleus and docked at NPCs at late times post infection. The data presented suggest that pUL16 and pUL21 synthesized after infection associate with NPCs thereby preventing the attachment of nascent cytoplasmic nucleocapsids.

pUL16 and pUL21 perform multiple functions in HSV infected cells, some independently and others as a complex. Both proteins appear to be more important for HSV-2 strains compared to HSV-1 strains in terms of the impacts of their deletion on virus replication. HSV-1

strains deleted for pUL16 have roughly 10-fold reductions in virus replication [33–35], whereas HSV-1 pUL21 deletion mutants have replication deficiencies in the range of 0 to 100-fold, depending on virus strain and cell type investigated [36–41]. Multiple studies have shown that passage of HSV-1 pUL21 deletion mutants on non-complementing cells results in the rapid selection of suppressor mutations that lead to enhanced virus replication and spread of infection between cells [37, 41]. These findings may suggest that previous analyses of pUL21 mutant strains that were not maintained on complementing cells [36, 39, 40] underestimated the impact of pUL21 deletion on HSV-1 replication. HSV-2 strains deleted for pUL16 showed 50 to 100-fold reductions in their replication [34, 42] and those deleted for pUL21 showed 50 to 1000-fold reductions [38, 43], depending on virus strain and cell type investigated. The requirements for pUL16 and pUL21 in the nuclear egress of HSV-2 strains, but not HSV-1 strains, may contribute to the differences in mutant virus replication between HSV species [34, 42–44]. Mutation of pUL16 or pUL21 in HSV-1 and HSV-2 results in a small plaque pheno-type on non-complementing cells indicating a role for these proteins in cell-to-cell spread of infection [34, 35, 37, 38, 41, 42]. A trimeric complex of pUL11, pUL16 and pUL21 assembles on the cytoplasmic tail of gE [45], a viral glycoprotein that plays an important role in cell-to-cell spread of infection [46]. This trimeric complex is critical for appropriate gE trafficking and function, likely explaining the small plaque phenotypes observed for pUL16 and pUL21 mutant strains [45]. pUL16 proteins are conserved throughout the *Herpesviridae* and are required for efficient secondary envelopment of capsids in viruses where they have been deleted [34, 35, 47–49]. pUL16 from HSV-1 has also been suggested to modulate mitochon-drial function. pUL16 contains sequences that recruit it to mitochondria [50] and, in addition, pUL16 binds the mitochondrial protein, adenine nucleotide transporter isoform 2, and pro-motes mitochondrial ATP synthesis [51]. Unlike pUL16, pUL21 expression is restricted to alphaherpesviruses [43]. HSV-1 pUL21 is an adaptor for protein phosphatase 1 that directs phosphatase activity to viral and cellular substrates [37, 52]. Both HSV-1 and HSV-2 pUL21 function to promote the dephosphorylation of pUL31 and pUL34, the principal components of the viral nuclear egress complex [53] that are phosphorylated by Us3, and in doing so regu-late the primary envelopment activity of these proteins [54–56]. Besides the aforementioned activities, pUL21 has been reported to interact with cytoskeletal components and to promote selective autophagy of innate immune system components [57, 58]. Here, we have revealed yet another function of HSV pUL16 and pUL21 in the prevention of nucleocapsid binding to NPCs thereby preventing a short circuit in virion assembly.

## Results

### Deletion of pUL16 or pUL21 results in capsid accumulation at the cytoplasmic face of the nuclear envelope late in infection

Vero cells were infected with HSV-2 strains containing mCherry (mCh) fused to the capsid protein VP26, stained for the nuclear lamina protein, lamin A/C, and DNA (Hoechst 33342), and examined by confocal microscopy at 18 hours post infection (hpi). Unlike what was observed in cells infected with the WT mCh-VP26 strain, cells infected with mCh-VP26 viruses deleted for pUL16 (Δ16), or pUL21 (Δ21), displayed an aberrant accumulation of mCh-VP26 fluorescence at the nuclear periphery (Fig 1A, arrowheads). Measurement of fluo-rescence intensities across the nuclear envelope demonstrated that, as expected, the peak DNA fluorescence intensity (Hoechst 33342) was inside the peak of nuclear lamina (lamin A/C) fluorescence (Fig 1B, 1C and 1D). Interestingly, the peak mCh-VP26 fluorescence intensity in Δ16 and Δ21 infected cells localized to the outside of the nuclear lamina suggesting that, unlike

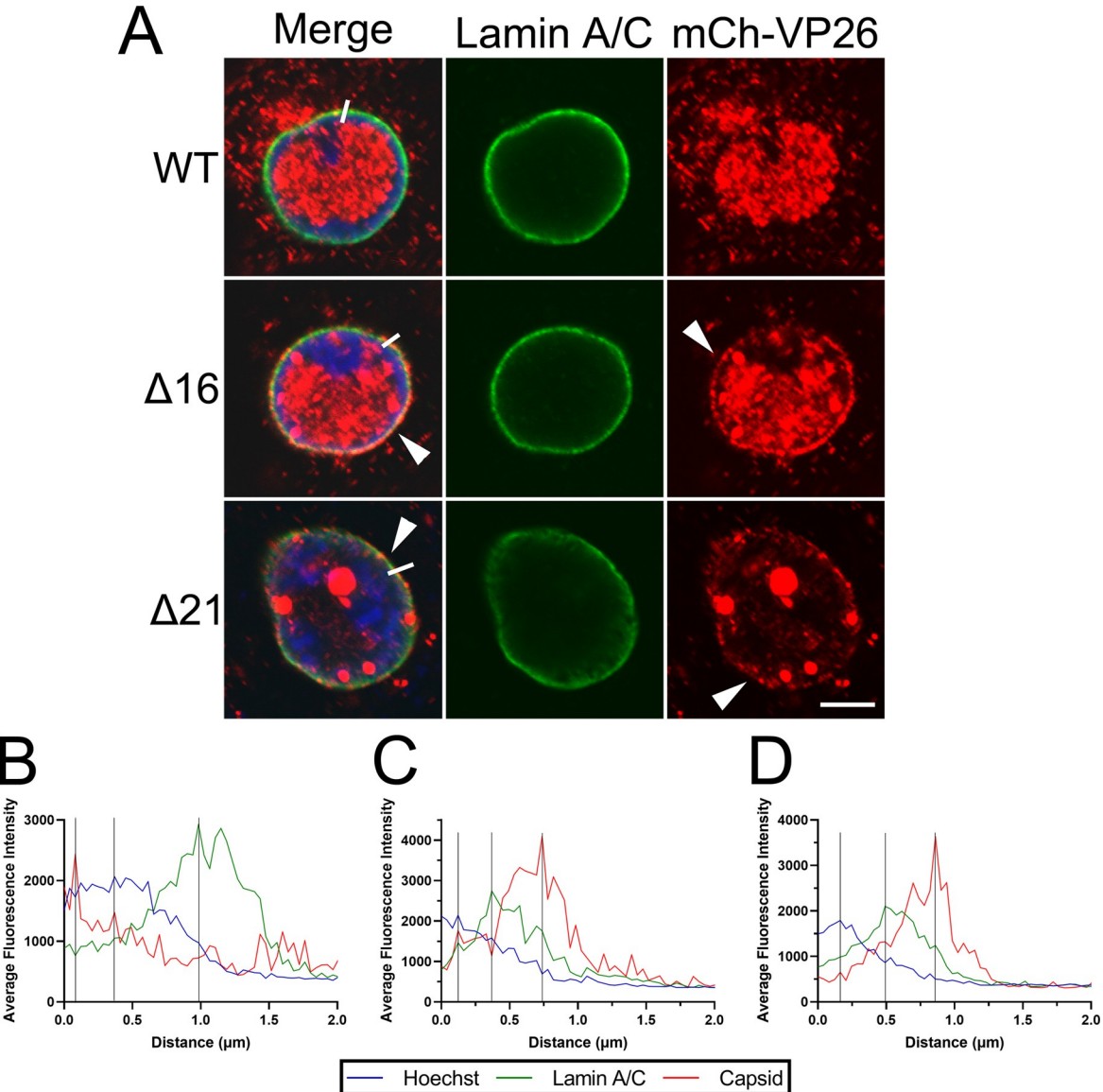

**Fig 1. The localization of HSV-2 186 mCh-VP26 WT, Δ16, and Δ21 capsids late in infection.** Vero cells were infected with HSV-2 186 mCh-VP26 WT, Δ16, or Δ21 virus at an MOI of 0.1 and were fixed at 18 hpi. After fixation, cells were permeabilized with TX-100 and stained for lamin A/C. **A)** Representative confocal images of HSV-2 infected cells. Merge shows the overlay of Hoechst 33342 (DNA) (blue), lamin A/C (green), and HSV-2 capsids (red). The scale bar indicates 5μm. Arrowheads indicate sites that displayed an aberrant accumulation of mCh-VP26 fluorescence at the nuclear periphery. The white lines indicate the location of fluorescence intensity measurements shown in B, C and D. **B**, **C** and **D)** Representative fluorescence intensity measurements for Hoechst, lamin A/C, and mCh-VP26 capsids in a WT (**B**) Δ16 (**C**) and Δ21 (**D**) infected cell starting within the nucleus (distance 0) and extending into the cytoplasm (n = 10 individual cells examined per condition). Peak intensities are indicated by the vertical black lines.

WT capsids (Fig 1B), Δ16 and Δ21 capsids were accumulating on the cytoplasmic face of the nuclear envelope (Fig 1C and 1D).

To verify that these observations were not due to the mCh-VP26 fusion altering capsid properties, we monitored capsid localization in cells infected with viruses expressing unfused VP26 using antisera against the HSV major capsid protein, VP5. Similar to what was seen using mCh-VP26 capsids, confocal microscopy of cells infected with HSV-2 Δ16, and Δ21

strains showed capsid accumulations at the nuclear periphery whereas cells infected with the HSV-2 WT strain did not (Fig 2A).

We next asked if the accumulation of Δ16 and Δ21 capsids at the nuclear periphery was restricted to HSV-2, or if it would also be applicable to HSV-1 mutants. Vero cells were infected with HSV-1 WT, Δ16, Δ21, or a pUL21/pUL16 double mutant (Δ21/Δ16), and capsids were visualized by staining for the HSV major capsid protein, VP5. Similar to what was observed with HSV-2 mutant strains, HSV-1 Δ16, Δ21, as well as Δ21/Δ16 infected cells, accumulated capsids at the nuclear periphery (Fig 2B).

To verify that capsid accumulation was occurring at the cytoplasmic face of the nuclear envelope rather than at its nucleoplasmic face, we utilized saponin to permeabilize infected cells prior to staining the cells with VP5 antisera. Saponin permeabilizes the plasma membrane but does not permeabilize the nuclear envelope [59], enabling VP5 antisera to bind cytoplasmic capsids, while nuclear capsids are not detected because they are inaccessible to the antisera. We frequently observed cytoplasmic capsids surrounding the nuclei of HSV-1 Δ16, Δ21, and Δ21/Δ16 infected cells whereas WT infected cells rarely displayed this capsid localization pattern (Fig 2C). Importantly, despite the abundance of capsids in the nuclei of virally infected cells (e.g., Fig 1A), no VP5 signal was evident within the nuclei of cells permeabilized with saponin, confirming that the nuclear envelope had not been breached in these cells. The percentage of cells with an abundance of capsids at the cytoplasmic face of the nuclear envelope was quantified. HSV-1 Δ16, Δ21, and Δ21/Δ16 infections had significantly more cells with capsids at the nuclear envelope in comparison to WT infected cells (Fig 2D). Interestingly, Δ21/Δ16 infected cells had significantly more cells with capsid accumulation at the nuclear envelope in comparison to Δ16 or Δ21 infected cells suggesting that the deletion of both pUL16 and pUL21 has an additive effect on capsid accumulation at the nuclear envelope.

## Capsids accumulating at the cytoplasmic face of the nuclear envelope colocalize with NPCs

Considering that cytoplasmic Δ16 and Δ21 capsids accumulate at the periphery of infected nuclei, we asked if these capsids colocalize with NPCs. To examine this, we infected Vero, life-extended human foreskin fibroblasts (T12), HeLa, and HaCaT cells with HSV-2 mCh-VP26 viruses, fixed cells at 18 hpi and stained for NPCs. In all cell types, HSV-2 mCh-VP26 Δ16 and Δ21 capsids, but not WT capsids, colocalized with NPCs (Figs 3A and S1). The colocalization of Δ16 and Δ21 capsids with NPCs was also assessed by measuring the fluorescence intensity of the capsids and NPCs across the nuclear envelope of infected Vero cells (Fig 3B and 3C). In this analysis, the peak of capsid fluorescence intensity and NPC fluorescence intensity coincided raising the possibility that capsids were docked at NPCs. As a complementary approach, HSV-1 WT, Δ16, Δ21 and Δ21/Δ16 infected Vero cells were analyzed by transmission electron microscopy (TEM). HSV-1 Δ16, Δ21, and Δ21/Δ16 capsids were commonly seen docked at NPCs at 18 hpi by TEM, but rarely in WT infected cells (Fig 3D and 3E). Additionally, capsids docked at NPCs late in infection uniformly lacked genomes suggesting that these capsids have ejected their genomes back into the infected nuclei. These findings indicate that pUL16 and pUL21 are important for preventing capsids from docking to NPCs at late stages of infection, irrespective of HSV species or cell type.

## Superinfection exclusion is impaired at the level of viral genome delivery in Δ16 and Δ21 infected cells

Similar to the findings described in Fig 3D, an analysis of the HSV-1 temperature-sensitive mutant, 50B, by Roizman and colleagues revealed the accumulation of empty capsids at the

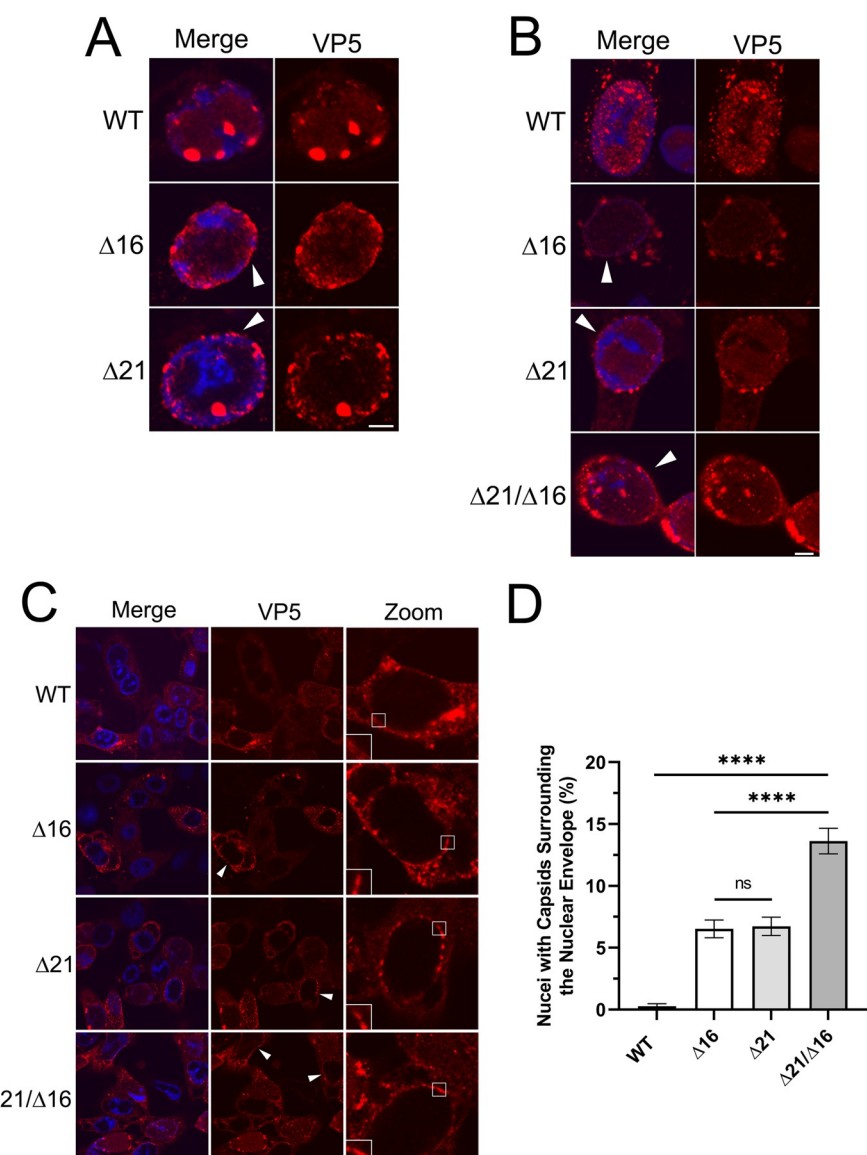

**Fig 2. The localization of HSV-2 and HSV-1 capsids in the absence of pUL16 and/or pUL21.** Vero cells were infected with either HSV-2 186 or HSV-1 KOS WT, Δ16, Δ21, or Δ21/Δ16 at an MOI of 1 and fixed at 18 hpi. After fixation, samples were permeabilized with TX-100, or saponin, and stained for the major capsid protein VP5. Merge in confocal images shows the overlay of Hoechst 33342 (blue) and HSV-2, or HSV-1, capsids (red). **A and B)** Representative confocal images of TX-100 permeabilized HSV-2 and HSV-1 infected cells, respectively, stained for VP5. Scale bars indicate 5μm. Arrowheads indicate nuclei with capsids at the nuclear envelope. **C)** Representative confocal images of saponin permeabilized HSV-1 infected cells stained for VP5. The scale bar in the lower magnification images is 20μm. The scale bar for zoom panels is 5μm. Arrowheads indicate nuclei with capsids at the nuclear envelope. Insets show the absence (WT panel) or presence (Δ16, Δ21, or Δ21/Δ16 mutant panels) of VP5 capsid fluorescence at the nuclear envelope. **D)** The quantification of the percentage of HSV-1 infected cells with capsids accumulated at the nuclear envelope. Three biological replicates with n = 198–258 infected cells examined per biological replicate. A one-way ANOVA with Tukey's HSD Test for multiple comparisons was performed between all viruses. **** represents p $\leq$ 0.0001.

cytoplasmic face of NPCs late in infection [60]. These authors suggested that the 50B strain bore mutations in viral glycoprotein genes that were responsible for conferring resistance of infected cells to superinfection by progeny virions; a process called superinfection exclusion

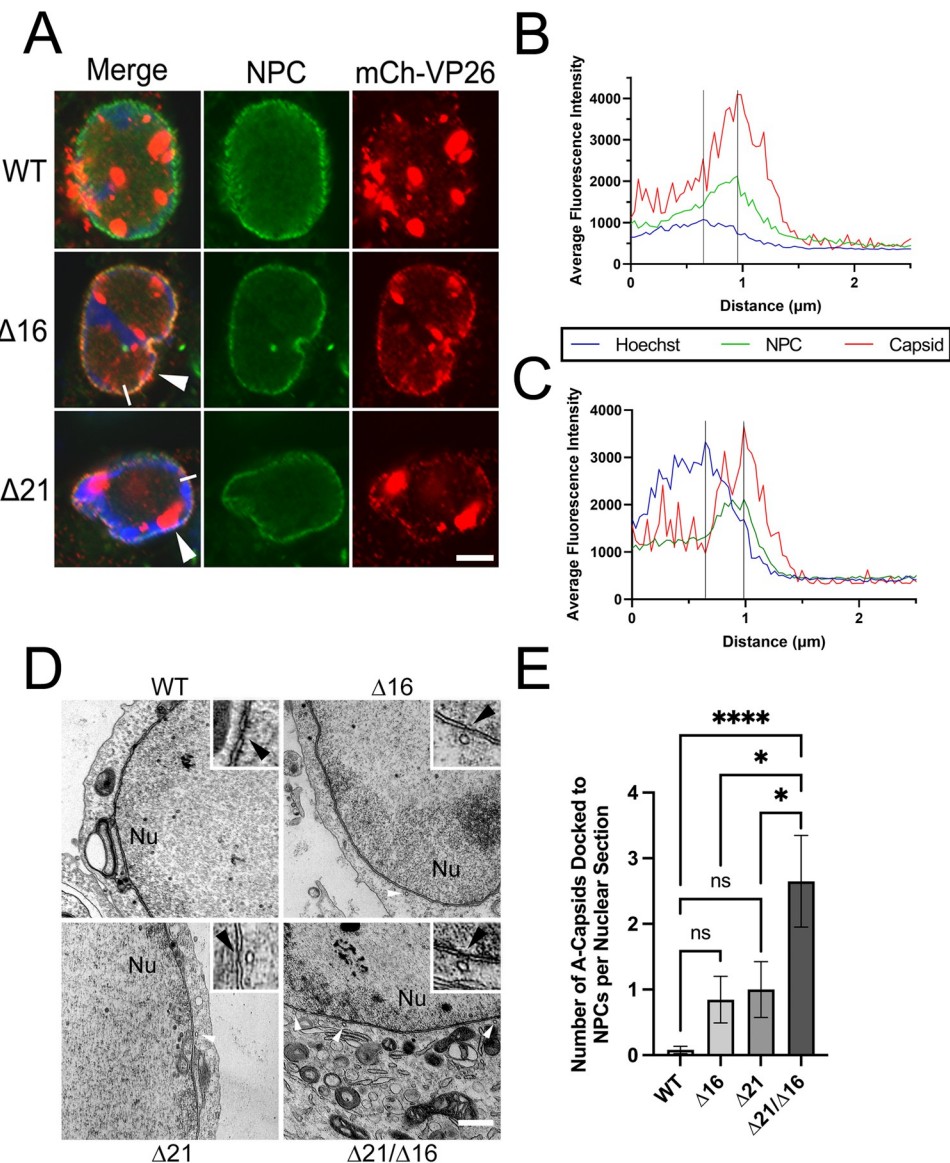

**Fig 3. The colocalization of HSV-2 capsids with NPCs at late times in infection.** Vero cells were infected with HSV-2 186 mCh-VP26 WT, Δ16, or Δ21 virus at an MOI of 0.1 and fixed at 18 hpi. After fixation, cells were permeabilized with TX-100 and stained for NPCs. **A)** Representative confocal images of HSV-2 infected Vero cells. Merge shows the overlay of Hoechst 33342 (DNA) (blue), NPCs (green), and HSV-2 capsids (red). The scale bar indicates 5μm. Arrowheads indicate cells with mCh-VP26 capsid fluorescence colocalized with NPC fluorescence. The white lines indicate the location of fluorescence intensity measurements shown in B and C. **B and C)** Representative fluorescence intensity measurements for Hoechst, NPCs, and HSV-2 capsids in a Δ16 and Δ21 infected Vero cell, respectively, starting within the nucleus (distance 0) and extending into the cytoplasm (n = 10 individual cells examined per condition). Peak intensities are indicated by the vertical black lines. **D)** TEM micrographs of capsids in HSV-1 infected cells. White arrowheads indicate capsids docked at NPCs and black arrowheads point to the nucleoplasmic face of NPCs in the zoomed inset panels. **E)** Quantification of the number of A-capsids docked at NPCs per nuclear section. n = 10–26 nuclear sections examined per condition. A one-way ANOVA with Tukey's HSD Test for multiple comparisons was performed between all viruses. * and **** represents $p \leq 0.05$ and $p \leq 0.0001$, respectively.

(SIE). Thus, the origin of the empty capsids accumulating at NPCs was suggested to be from re-infection of cells by progeny virions, leading to the delivery of capsids to NPCs and ejection of their genomes into the infected cell nucleoplasm.

To assess whether SIE was altered in cells infected with pUL16 and pUL21 mutant strains, we measured the entry of a second, superinfecting virus and the delivery of capsids to the nuclear envelope. Vero cells were mock infected or infected with HSV-2 186 WT, Δ16, or Δ21 viruses. At 6 hpi, cells were superinfected with WT HSV-2 186 mCh-VP26 virus. One hour after superinfection, cells were stained with fluorescent phalloidin to visualize actin at the cell periphery and the entry of mCh-VP26 capsids into cells was quantified by confocal microscopy (Fig 4A). Importantly, when HSV-2 186 mCh-VP26 virus was absorbed to cells on ice, immediately fixed, and stained with phalloidin, the majority of mCh-VP26 capsids were observed at the cell surface (S2A and S2B Fig). HSV-2 infected cells had significantly fewer cytoplasmic superinfecting mCh-VP26 capsids compared to the mock infected cells (Fig 4B). These findings suggest that HSV-2 infected cells display some degree of SIE at the point of viral entry; however, there was no significant difference in the numbers of mCh-VP26 capsids in the cytoplasm of Δ16 and Δ21 infected cells compared to WT infected cells. This suggests that Δ16 and Δ21 infected cells inhibit entry of a superinfecting virus comparably to WT infected cells. Additionally, the localization of superinfecting capsids to the nuclear periphery was similar in all conditions tested, suggesting that the transport of capsids from the cell periphery towards the nucleus was unaffected (Fig 4C).

Next, we measured the efficiency with which superinfecting capsids could dock to NPCs and eject their genomes into the nucleoplasm of infected cells. To do this, Vero cells were first infected with HSV-2 186 WT, Δ16, or Δ21 strains and at 6 hpi were superinfected with HSV-2 186 WT virus that had the nucleoside analog, 5-ethynyl-2'-deoxycytidine (EdC), incorporated into its genome. Three hours after superinfection, click-chemistry was performed to label the EdC with AlexaFluor-488 to enable the visualization of viral genomes that had been delivered to the nucleoplasm by superinfecting virions. Interestingly, Δ16 and Δ21 infected cells had significantly more nuclear genomes from the superinfecting virions compared to cells that had been initially infected with the WT strain (Fig 4D and 4E). Importantly, when HSV-2 186 WT EdC virus was absorbed to cells at 4˚C, immediately fixed, and click-chemistry was performed, no nuclear EdC-labelled viral genomes were observed (S2C Fig). Consistent with the results shown above, these data suggest that the expression of pUL16 and pUL21 during infection prevents genome ejection into infected nuclei from superinfecting nucleocapsids.

## Analysis of pUL16 and pUL21 prevention of capsid recruitment to NPCs

We propose three mechanisms that could explain how pUL16 and pUL21 prevent capsid docking to NPCs. First, pUL16 and/or pUL21 bound to cytoplasmic capsids may mask capsid proteins responsible for NPC docking. Second, pUL16 and/or pUL21 may influence capsid interactions with microtubules and/or microtubule motor proteins that enable their efficient transport away from the nuclear periphery towards sites of secondary envelopment. Finally, pUL16 and/or pUL21 may bind to, or alter, NPCs to prevent capsid docking.

During HSV infection, pUL16 and pUL21 are not expressed until between 4 and 6 hpi when nuclear egress and capsid transport to sites of secondary envelopment are also occurring [61]. Since incoming capsids dock at NPCs before new pUL16 and pUL21 are synthesized, we determined if expression of pUL16 and pUL21 prior to infection blocked incoming capsids from being recruited to the nuclear envelope. HeLa and PK15 cells were transfected with HSV-2 pUL16 or pUL21 expression plasmids, alone or in combination. At 24 hours post transfection, the recruitment of incoming HSV-2 or pseudorabies virus (PRV) capsids to the nuclear periphery was monitored by measuring the distance between mCh-VP26 capsids and the perimeter of the Hoechst 33342 signal (Fig 5A–5C). In these experiments, co-transfection of an EGFP expression plasmid enabled the identification of transfected cells. Interestingly,

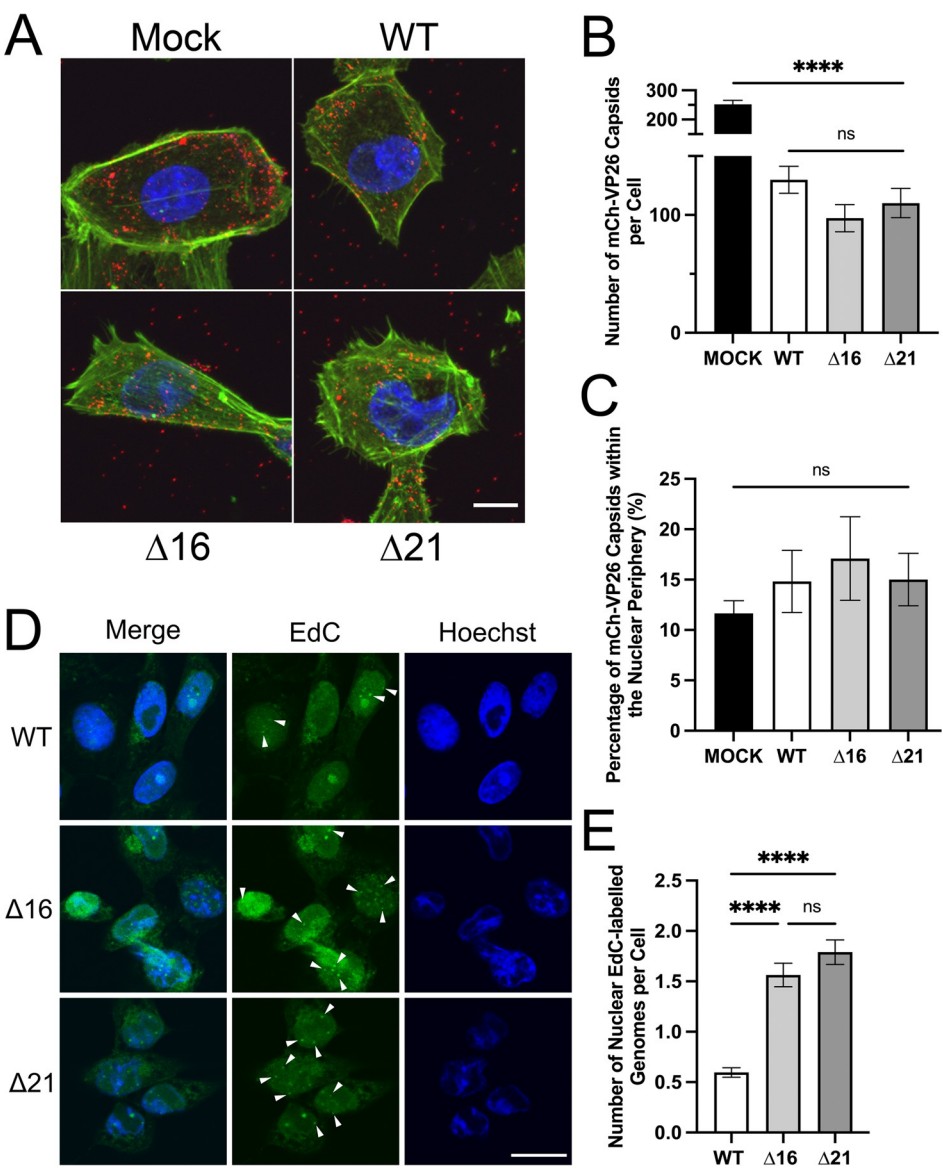

**Fig 4. Examination of SIE in HSV-2 WT, Δ16 and Δ21 infected cells. A)** Representative z-projections of cells that were initially mock infected or infected with HSV-2 186 WT, Δ16 or Δ21 virus at an MOI of 1 and at 6 hpi superinfected with HSV-2 186 mCh-VP26 at an MOI of 3 for one hour. z-projections show the overlay of Hoechst 33342 (DNA) (blue), phalloidin staining of actin filaments (green) and HSV-2 186 mCh-VP26 capsids (red). The scale bar is 10μm. **B)** Quantification of cytoplasmic mCh-VP26 capsids within mock, WT, Δ16 and Δ21 infected cells. n = 10 z-projections of infected cells per condition. **C)** Quantification of the percentage of mCh-VP26 capsids within the nuclear periphery of mock, WT, Δ16 and Δ21 infected cells. n = 10 z-projections of infected cells per condition. Capsids at the nuclear periphery were defined as those abutting the Hoechst 33342 signal. **D)** Representative confocal images of HSV-2 186 WT, Δ16 and Δ21 infected cells infected with HSV-2 186 WT EdC labelled virus at 6 hpi and fixed at 3h after WT EdC infection. Fluorescent puncta from WT EdC ejected genomes are seen in the nuclei of WT, Δ16 and Δ21 infected cells as indicated by the arrowheads in the figure. The scale bar indicates 20μm. **E)** Quantification of nuclear WT EdC labelled genomes within WT, Δ16 and Δ21 infected cells. n = 177–470 infected cells examined for fluorescent EdC puncta per biological replicate. One-way ANOVAs with Tukey's HSD Test for multiple comparisons were performed between all viruses. **** represents p ≤ 0.0001.

expression of pUL16 or pUL21 alone did not significantly affect the average distance of incoming HSV-2 or PRV capsids from the nuclear periphery in comparison to control cells that had been transfected with an EGFP expression plasmid alone (Fig 5B and 5C). Only the co-

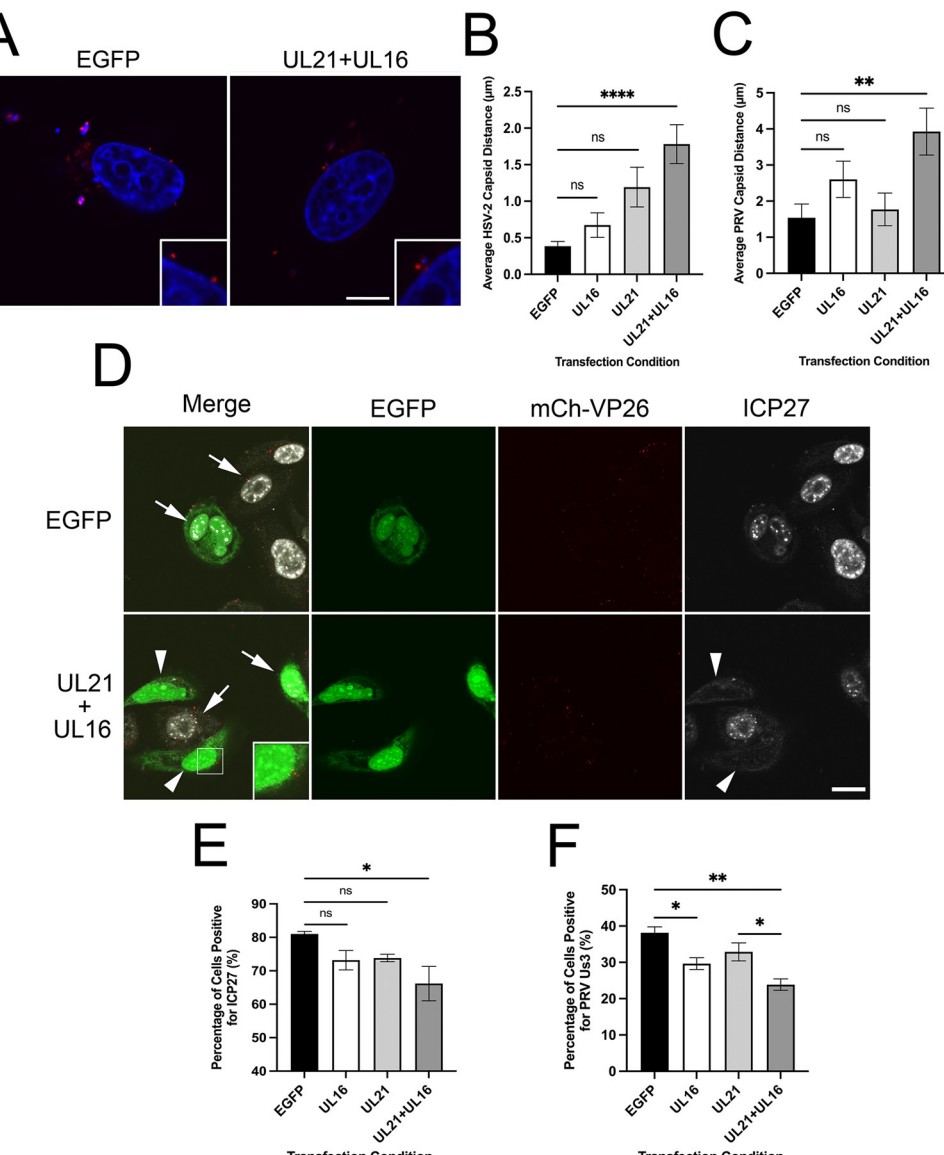

**Fig 5. The effect of prior ectopic HSV-2 pUL16 and pUL21 expression on HSV-2 and PRV capsid recruitment to nuclei and subsequent virus gene expression. A)** Representative images of HeLa cells transfected with EGFP (control) or HSV-2 pUL16 and pUL21 expression plasmids 24 hours prior to infection with HSV-2 186 mCh-VP26. Cells were imaged at 2 hpi. Hoechst 33342 (DNA) (blue), HSV-2 186 mCh-VP26 capsids (red). Scale bar is 10μm. **B)** Average distance of HSV-2 capsids from the nuclear periphery in HeLa cells that had been transfected with EGFP alone (control) or in combination with HSV-2 pUL16 and/or pUL21 expression plasmids and infected for 2h with HSV-2 186 mCh-VP26. n = 43–55 capsids were examined in each transfection condition. A one-way ANOVA with Tukey's HSD Test for multiple comparisons was performed between all transfection conditions. **** represents p ≤ 0.0001. **C)** Average distance of PRV capsids from the nuclear periphery in PK15 cells that had been transfected with EGFP alone (control) or in combination with HSV-2 pUL16 and/or pUL21 expression plasmids and infected for 2h with PRV765. A one-way ANOVA with Tukey's HSD Test for multiple comparisons was performed between all transfection conditions. **** represents p ≤ 0.0001. n = 46–60 capsids were examined in each transfection condition. **D)** Confocal images showing ICP27 expression in cells transfected with EGFP alone (control) or in combination with HSV-2 pUL16 and pUL21 expression plasmids and infected for 2h with HSV-2 186 mCh-VP26. Arrows indicate cells expressing ICP27. Arrowheads indicate transfected and infected cells lacking ICP27 expression (mCh signal). The inset image shows mCh-VP26 capsids inside a cell, that does not express ICP27 and has been co-transfected with pUL16 and pUL21 expression plasmids. The scale bar is 20μm. **E and F)** Quantification of the percentage of transfected cells positive for HSV-2 ICP27 (**E**) or PRV Us3 expression (**F**) at 2 hpi. n = 40–110 transfected cells examined per biological replicate. Cells positive for ICP27 and Us3 expression were scored by visualization directly under the microscope. A one-way ANOVA with Tukey's HSD Test for multiple comparisons was performed between all transfection conditions. *, **, and **** represent p ≤ 0.05, p ≤ 0.01, and p ≤ 0.0001, respectively.

expression of pUL16 and pUL21 significantly increased the average distance of HSV-2 and PRV capsids from the nuclear periphery raising the possibility that a pUL16/pUL21 complex may be required to prevent capsid recruitment.

As a complementary approach, viral protein expression was examined to determine if incoming capsids were successfully delivering their genomes into nuclei when pUL16 and/or pUL21 were expressed prior to infection. Only the co-expression of pUL16 and pUL21 significantly decreased the number of cells producing the HSV-2 immediate early protein ICP27 in comparison to the EGFP control (Fig 5D and 5E). Importantly, mCh-VP26 capsids were visible in the cytoplasm of co-transfected cells lacking ICP27 expression indicating that virus had entered these cells (Fig 5D, inset). Ectopic expression of pUL16 alone and in combination with pUL21 significantly decreased the number of cells expressing the PRV Us3 protein in comparison to the EGFP control (Fig 5F). These data suggest that co-expression of both pUL16 and pUL21 prior to infection effectively prevents capsids from docking at NPCs and the subsequent delivery of viral genomes to the nucleus. Because co-expression of HSV-2 pUL16 and pUL21 also disrupted the recruitment of the distantly related PRV capsids to the nuclear periphery, the data suggest that ectopic expression of pUL16 and pUL21 interfere with nuclear envelope factors that influence capsid recruitment rather than blocking NPC-interaction surfaces on capsids.

To explore this idea further, we investigated if ectopically expressed pUL16-EGFP and pUL21-EGFP could be recruited to incoming capsids. HeLa cells were transfected with combinations of plasmids encoding pUL16, pUL21 and their EGFP fusions. At 24 hours post transfection, cells were infected with HSV-2 WT mCh-VP26, and the EGFP signal associated with mCh-VP26 capsids at 2 hpi was evaluated by confocal microscopy (Fig 6A and 6B). No significant differences in capsid-associated EGFP signal were observed in cells transfected with pUL16-EGFP, pUL21-EGFP or EGFP expression plasmids (Fig 6B). Additionally, we noted that when pUL16-EGFP was expressed alone, it had a pancellular distribution; however, when co-expressed with pUL21, a fraction of pUL16-EGFP also localized to the nuclear envelope (Fig 6A, inset). This finding further supports the idea that pUL16 and pUL21 function together at the nuclear envelope to prevent capsid recruitment.

To confirm that the pUL16 and pUL21 EGFP fusion proteins had the capacity to interact with capsids, we showed that ectopically expressed pUL16-EGFP and pUL21-EGFP colocalized with mCh-VP26 capsids at late times post infection (Fig 6C and 6D). In agreement with previous reports, pUL21 was associated with capsids in both the nucleus and cytoplasm, whereas pUL16 was associated with cytoplasmic capsids but not nuclear capsids [58, 62–64] (Fig 6C and 6D). Consistent with experiments shown above, Δ16 and Δ21 capsids colocalized with NPCs only in EGFP transfected cells (Fig 6C and 6D, grey arrowheads). Collectively, the data shown in Fig 6 suggest that pUL16 and pUL21 are not loaded onto incoming capsids when ectopically expressed and it is, therefore, unlikely that the binding of pUL16 and pUL21 to capsids prevents their recruitment to the nuclear envelope.

After nuclear egress, cytoplasmic capsids are transported away from the nuclear periphery to membranes that serve as sites of secondary envelopment. HSV-1 and PRV pUL21 interact with components of the microtubule-transport machinery raising the possibility that pUL21, or a pUL16/pUL21 complex, may enhance nascent capsid transport [58, 65]. Thus, we hypothesized that in the absence of pUL16 and pUL21, nascent cytoplasmic capsids might linger at the nuclear periphery, providing a greater opportunity for these capsids to dock at NPCs. If this were the case, we predicted that disruption of microtubule-based transport with nocodazole might enable nascent WT capsids to dock at NPCs.

Nocodazole reversibly interferes with microtubule polymerization [66]. Treating cells with this agent disrupted the microtubule network (Fig 7A) and prevented the delivery of incoming

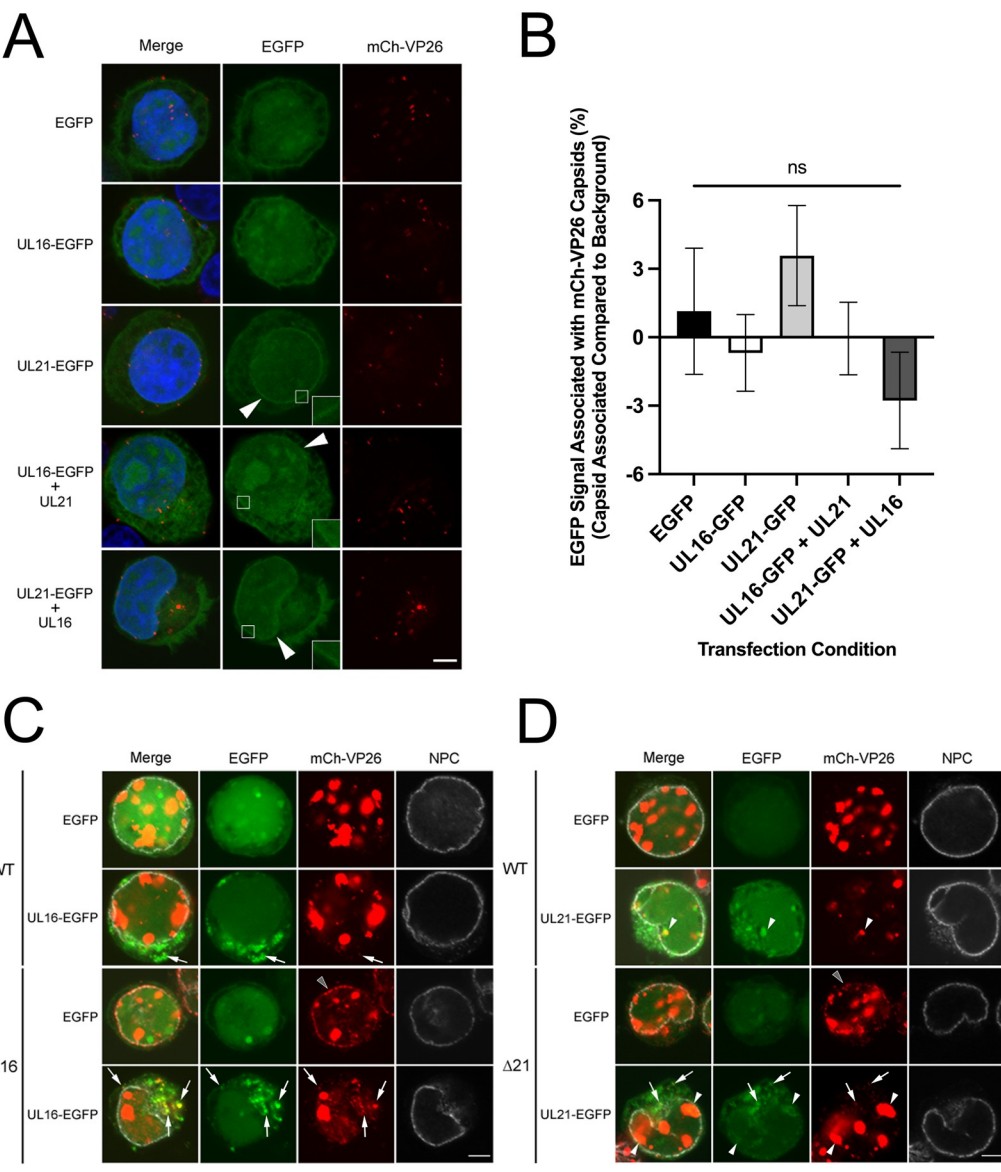

**Fig 6. The colocalization of pUL16-EGFP and pUL21-EGFP with HSV-2 mCh-VP26 capsids. A)** HeLa cells were transfected with EGFP, pUL16-EGFP, pUL21-EGFP, pUL16-EGFP and pUL21, or pUL21-EGFP and pUL16 expression plasmids. At 24 hours post transfection, transfected cells were infected on ice with HSV-2 186 mCh-VP26 at a MOI of 3 then shifted to 37°C and fixed at 2 hpi. Scale bar indicates 5µm. Arrowheads indicate cells with pUL16-EGFP/pUL21-EGFP nuclear rim staining. **B)** Colocalization between EGFP signal and mCh-VP26 capsid signals was compared to the average EGFP background signal taken from two points 0.5µm from the analyzed capsid. n = 40 capsids per transfection condition were examined. A one-way ANOVA with Tukey's HSD Test for multiple comparisons was performed between all transfection conditions. No significant differences were seen between any of the transfection conditions when comparing the EGFP signal associated with mCh-VP26 capsids to the EGFP control. **C and D)** HeLa cells were transfected with EGFP, pUL16-EGFP or pUL21-EGFP expression plasmids. At 24 hours post transfection, cells were infected with HSV-2 186 mCh-VP26 WT, Δ16 or Δ21 virus at a MOI of 3 and fixed at 18 hpi. Fixed cells were permeabilized and stained for NPCs. Δ16 and Δ21 capsids colocalized with NPCs only in EGFP transfected cells (grey arrowheads). White arrowheads and arrows depict EGFP signal colocalized with mCh-VP26 capsid signal in the nucleus and cytoplasm, respectively. Scale bars indicate 5µm.

HSV capsids to NPCs and subsequent viral gene expression (Fig 7B). Cells were infected with WT, Δ21, or Δ16 HSV-2 strains bearing mCh-VP26 capsids and at 6 or 12 hpi, treated with nocodazole or vehicle (DMSO) until 18 hpi. Cells were stained for NPCs and colocalization

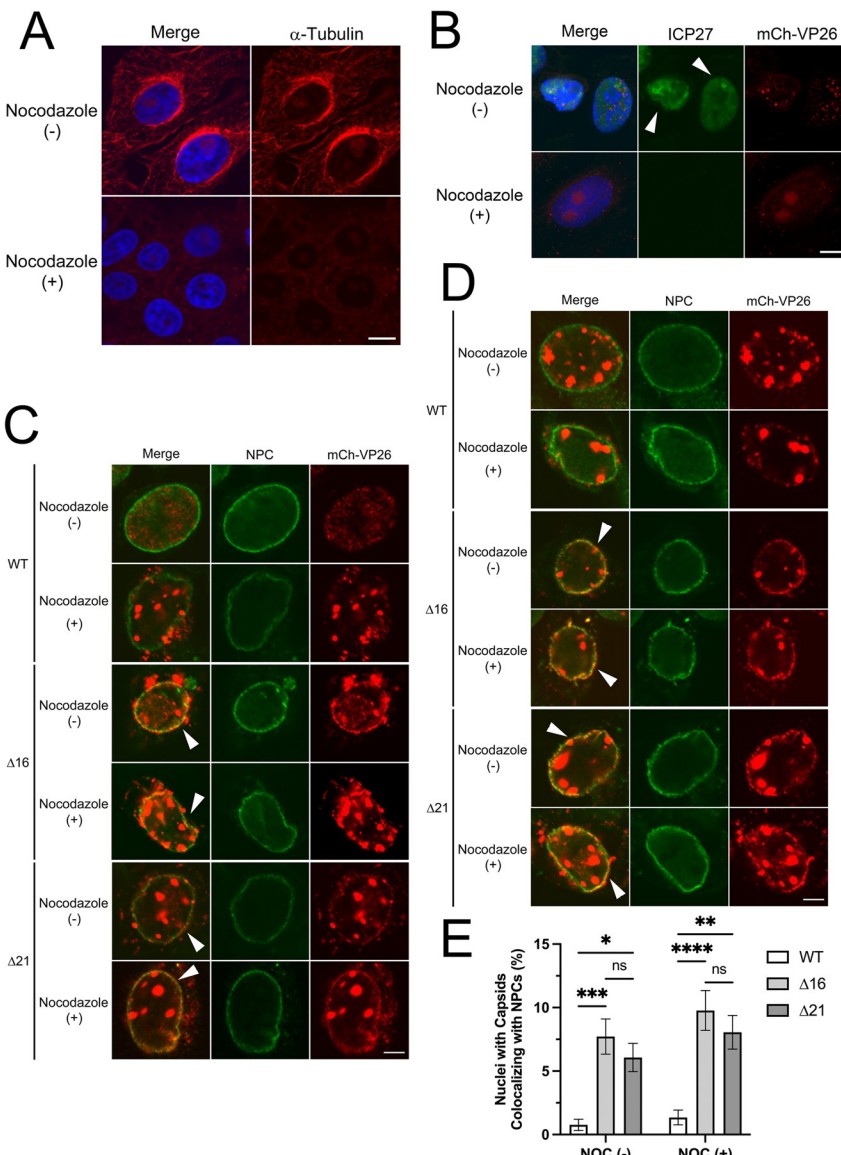

**Fig 7. The colocalization of HSV capsids with NPCs in cells treated with nocodazole. A)** Representative confocal images of mock infected cells treated with DMSO or 10μM nocodazole for 6 hours. Merge shows the overlay of Hoechst 33342 (DNA) (blue) and α-tubulin (red). The scale bar indicates 10μm. **B)** Representative confocal images of cells treated with DMSO or 10μM nocodazole for 30 mins prior to infection on ice for 1 hour at an MOI of 3. Samples were shifted to 37˚C and fixed at 3 hpi. Merge shows the overlay of Hoechst 33342 (DNA) (blue), ICP27 (green) and mCh-VP26 capsids (red). Arrowheads indicate cells with nuclear ICP27 expression. As seen in the nocodazole-treated cells, expression of ICP27 is absent as capsid transport to the nucleus is prevented. The scale bar indicates 10μm. **C and D)** Representative confocal images of HSV-2 infected cells treated with DMSO or nocodazole at 6 hpi or 12 hpi, respectively and fixed at 18 hpi. Merge shows the overlay of NPCs (green) and HSV-2 mCh-VP26 capsids (red). Scale bars indicate 5μm. Arrowheads indicate cells with mCh-VP26 capsid signal colocalized with NPC signal. **E)** Quantification of nuclei with capsids colocalizing with NPCs in HSV-2 infected cells treated with DMSO or nocodazole at 12 hpi and fixed at 18 hpi. Three biological replicates with n = 100–140 infected cells examined per biological replicate. A two-way ANOVA with Tukey's HSD Test for multiple comparisons was performed between all viruses and treatment conditions. *, **, *** and **** represent p ≤ 0.05, p ≤ 0.01, p ≤ 0.001 and p ≤ 0.0001, respectively.

between mCh-VP26 capsids and NPCs assessed by confocal microscopy (Fig 7C and 7D). Whereas mCh-VP26 capsids colocalized with NPCs in Δ16 and Δ21 infected cells, WT infected cells showed limited colocalization of capsids with NPCs either in the presence or absence of nocodazole suggesting that microtubule de-polymerization does not enhance nascent cytoplasmic capsid recruitment to NPCs (Fig 7C–7E). Importantly, the efficient colocalization of Δ21 and Δ16 mCh-VP26 capsids with NPCs in the presence of nocodazole suggests that the origin of these capsids was the infected cell nucleus rather than from superinfecting virions that would neither be efficiently released from cells, nor be able to reach the nuclear envelope in the absence of a microtubule network.

The data so far suggest that the expression of pUL16 and pUL21 prior to infection prevents incoming capsids from docking to NPCs (Fig 5), and that when pUL16 and pUL21 are co-expressed, they localize to the nuclear envelope (Fig 6A). Additionally, previous work from our own and other laboratories demonstrated that a population of pUL21 exhibits a nuclear rim like localization in HSV-1 and HSV-2 infected cells [37, 43]. Considering this, we wanted to explore if pUL16 and pUL21 localize to the cytoplasmic face of nuclei, placing them in an appropriate location to interfere with nucleocapsid/NPC interactions. Vero cells were infected with an HSV-2 186 recombinant virus expressing mCh fused to the carboxy terminus of pUL21 (pUL21-mCh) for 18 h. Stimulated emission depletion (STED) microscopy was utilized to examine the localization of pUL21-mCh with respect to lamin A/C and RanBP2 to further understand pUL21 localization within the nuclear envelope. Lamin A/C is a component of the nuclear lamina and found adjacent to the inner nuclear membrane. RanBP2 forms the NPC filaments found on the cytoplasmic face of NPCs and is important for HSV nucleocapsid docking to NPCs [17, 20]. Cross sections through the centre of an infected cell nucleus showed pUL21-mCh in the same plane as lamin A/C and RanBP2; suggesting that pUL21-mCh localizes to both the nuclear and cytoplasmic faces of the nuclear envelope (Fig 8A and 8B, cross section). When the top of the infected cell nucleus was examined to visualize pUL21-mCh in respect to RanBP2, pUL21-mCh was colocalized (arrowheads) and adjacent to RanBP2, however, the majority of pUL21-mCh was seen in a lattice-like distribution on the nuclear envelope (Fig 8B, top). Interestingly, the spaces between pUL21-mCh signals were 120nm to 180nm in size, consistent with the size of NPCs (~120nm in diameter) [26] (Fig 8B, 10X zoom). As a complementary approach, we investigated the localization of pUL21-mCh in relation to other NPC components by staining infected cells with MAb414 that reacts with the FG-repeats found in several nucleoporins [67]. Similar to the findings in RanBP2 stained cells, pUL21-mCh localized in a lattice-like distribution in the nuclear envelope and was localized adjacent to, and with (arrowheads), NPC components in both cross section and at the nuclear surface (Fig 8C). These data suggest that, during infection, pUL21 is localized to the nuclear envelope and is in proximity to NPC components.

Since a fraction of pUL16 localized to the nuclear envelope by confocal microscopy, when pUL21 was co-expressed (Fig 6A), we investigated the localization of pUL16 within the nuclear envelope at higher resolution by STED microscopy. Because our pUL16 antisera does not work in fluorescence microscopy applications, we examined pUL16-mCh localization when co-expressed with pUL21 in transfected cells. Vero cells were transfected with pUL16-mCh and pUL21, or with pUL21-mCh and pUL16 expression plasmids. At 24 hours post transfection, samples were fixed and stained for mCh and NPCs (MAb414). Similar to pUL21 localization in infected cells (Fig 8), pUL16-mCh localized in the same plane as NPCs and in a lattice-like distribution in the nuclear envelope (Fig 9A). Additionally, pUL16-mCh was seen adjacent to and colocalized (arrowheads) with NPC components (Fig 9A). pUL16-mCh localization in respect to lamin A/C was also examined (S3 Fig). A fraction of pUL16 localized in the same plane as lamin A/C, suggesting that pUL16 also localizes to the nucleoplasmic face of the

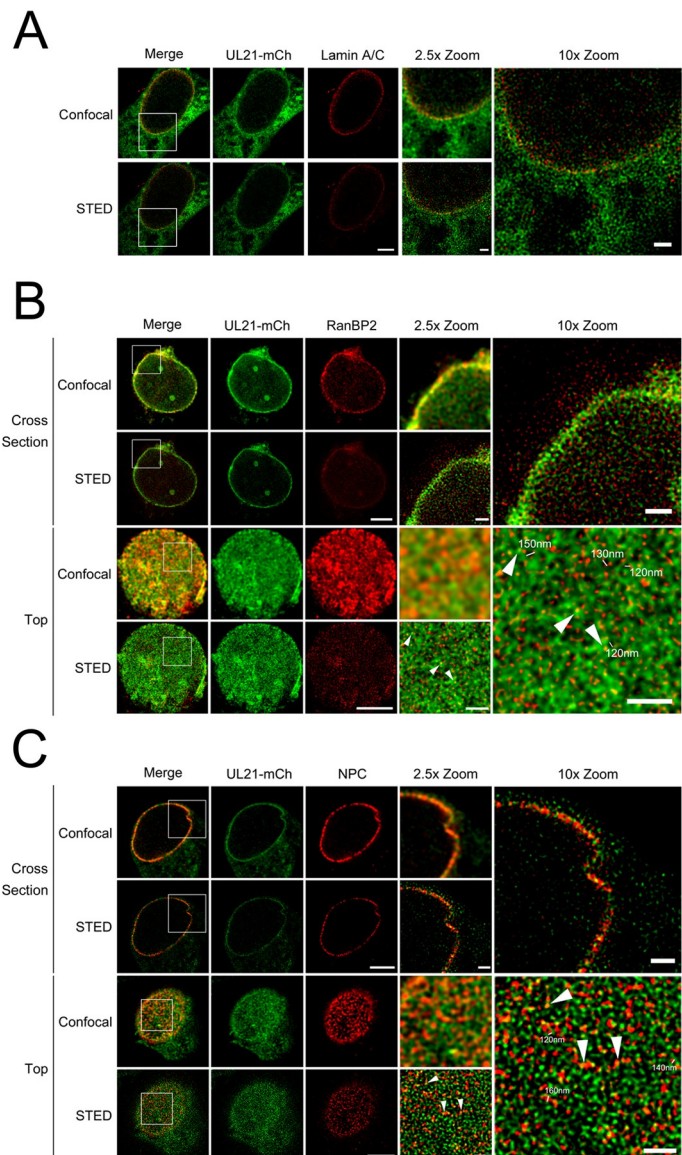

**Fig 8. pUL21 localization during infection.** Cells were imaged by confocal microscopy and stimulated emission depletion (STED) microscopy. **A)** Cross section of an HSV-2 186 pUL21-mCh infected nuclei stained with antisera against mCh and lamin A/C is shown. **B and C)** Cross section and top views of HSV-2 186 pUL21-mCh infected nuclei stained with antisera against mCh and RanBP2 (**B**) or MAb414 (**C**) that reacts with multiple NPC components. The 2.5X and 10X zoomed regions are indicated in the merge panels. Scale bars in the lower magnification images are 5μm. Scale bars in the 2.5X and 10X zoom images are 1μm. Arrowheads indicate pUL21-mCh signal colocalized with RanBP2 or NPC signal. Measurements in 10X zoom panels indicate distances between pUL21-mCh signals.

nuclear envelope (S3 Fig). Importantly, when pUL16-mCh was expressed in cells in the absence of pUL21, pUL16 did not localize to the nuclear envelope and had a pancellular distribution (S3 Fig). In cells co-expressing pUL21-mCh and pUL16, pUL21 was seen adjacent to and colocalized with NPC components in a manner indistinguishable from what was observed in infected cells (Figs 9B and 8C). Collectively, these data suggest that both pUL16 and pUL21 are in proximity to NPCs and could interfere with NPC components that mediate capsid docking. A model describing how pUL16 and pUL21 might prevent nascent cytoplasmic nucleocapsids from engaging NPCs is shown in Fig 10.

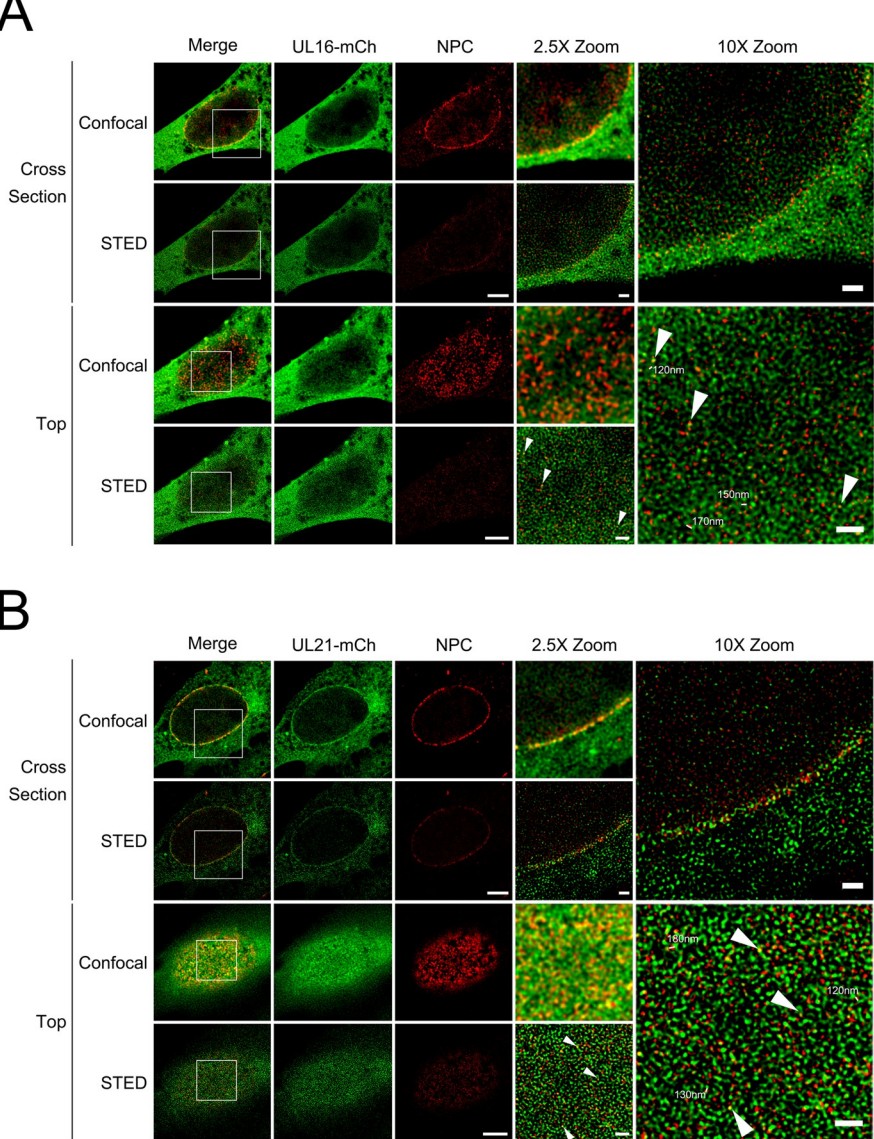

**Fig 9. pUL16 and pUL21 localization in transfected cells.** Cells were imaged by confocal microscopy and stimulated emission depletion (STED) microscopy. Cross section and top views of nuclei co-transfected with pUL16-mCh and pUL21 expression plasmids (**A**) or pUL21-mCh and pUL16 expression plasmids (**B**). Cells were stained with antisera against mCh and MAb414 that reacts with multiple NPC components. The 2.5X and 10X zoomed regions are indicated in the merge panels. Scale bars in the lower magnification images are 5μm. Scale bars in the 2.5X and 10X zoom images are 1μm. Arrowheads indicate pUL16-mCh (**A**) and pUL21-mCh (**B**) signal colocalized with the NPC signal. Measurements in 10X zoom panels indicate distances between pUL16-mCh/pUL21-mCh signals.

## Discussion

Despite HSV infecting many hundreds of millions of persons worldwide and more than half a century of intensive research into these important human pathogens, many questions related to fundamental aspects of HSV biology remain unanswered. It is well understood that after entry into the cell, incoming HSV capsids dock at NPCs to release their genomes into the host cell nucleoplasm (Fig 10A) [15–17]. However, why capsids egressing from the nucleus fail to interact with NPCs is poorly understood. Here, we have demonstrated that pUL16 and pUL21

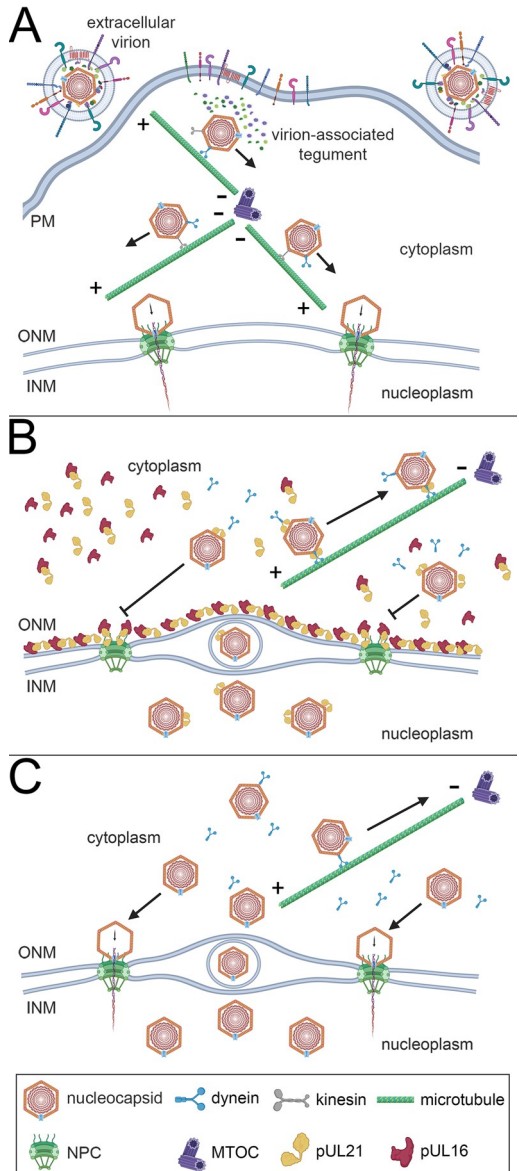

**Fig 10. Model describing the role of pUL16 and pUL21 in preventing nascent nucleocapsid recruitment to NPCs.**
**A)** To initiate infection, the virion envelope fuses with a cellular membrane, such as the plasma membrane (PM), and the nucleocapsid and virion-associated tegument are released into the cytoplasm. Most of the virion-associated tegument dissociates from the nucleocapsid and the capsid is transported along microtubules towards the microtubule organizing center (MTOC) using the minus-end directed motor protein dynein. After reaching the MTOC, nucleocapsid transport to the nucleus is facilitated by a plus-end directed kinesin motor protein family member. Once at the nuclear periphery, nucleocapsids dock at NPCs, triggering the delivery of the viral genome into the nucleoplasm. **B)** During virion assembly, nucleocapsids transit from the nucleoplasm to the cytoplasm by acquiring a primary envelope at the inner nuclear membrane (INM) followed by fusion of this primary envelope with the outer nuclear membrane (ONM). Once in the cytoplasm, nucleocapsids are transported away from the nuclear periphery towards the MTOC using dynein. Nucleocapsid associated pUL21, acquired during capsid maturation in the nucleoplasm, may aid the recruitment of dynein to the nucleocapsid. pUL16 and pUL21 form a lattice-like array on the cytoplasmic face of the ONM where these proteins are adjacent to, and co-localized with, NPC components. We hypothesize that the localization of pUL16 and pUL21 to the cytoplasmic face of the ONM prevents nascent cytoplasmic nucleocapsids from docking to NPCs and ejecting their genomes into the nucleoplasm. **C)** In cells infected with mutant viruses that do not express pUL16 and/or pUL21 (the double-mutant scenario is shown), the cytoplasmic face of the ONM and NPC are not modified through association with pUL16 and pUL21 and attachment of nascent cytoplasmic nucleocapsids to NPCs is not prevented, nor is the delivery of their genomes to the infected cell nucleus. Created with BioRender.com.

are required to prevent both HSV-1 and HSV-2 nascent cytoplasmic nucleocapsids from docking to NPCs (Figs 1–3) and that ectopic co-expression of pUL16 and pUL21 prior to infection interferes with both HSV-2 and PRV capsid delivery to the nuclear envelope and prevents initiation of virus gene expression (Fig 5). Additionally, we found that co-expression of pUL21 was required to recruit pUL16 to the nuclear envelope where interference with capsid/NPC interactions might be expected to take place (Figs 6A and 9A). These findings suggest that both pUL16 and pUL21 are required to prevent capsids from docking to NPCs. Considering these proteins are binding partners [31, 32], it may be that the pUL16/pUL21 complex is required for this activity. Graham and co-workers have demonstrated that mutation of HSV-1 pUL21 lysine 490 to glutamic acid interferes with its interaction with pUL16 [52]. Analysis of virus strains that have been engineered to express pUL16 or pUL21 mutants that fail to interact with each other, while beyond the scope of the present study, should provide a definitive answer to this question.

We hypothesized that differences in capsid composition between incoming and egressing capsids may prevent egressing capsids from interacting with NPCs. During entry, the majority of the tegument dissociates from the capsid [3–7]; however, during egress, tegument proteins are loaded onto capsids [30]. It may be that capsid-associated pUL16 and pUL21 on egressing nucleocapsids mask capsid proteins, such as pUL25 and pUL36, that function in capsid recruitment to NPCs. We were unable to detect pUL16 and pUL21 binding to incoming capsids when pUL16 and pUL21 were ectopically expressed prior to infection (Fig 5), suggesting that pUL16 and pUL21 interference with capsid/NPC interactions does not involve pUL16 or pUL21 binding to capsids. However, a limitation of these experiments is that undetectable levels of pUL16 and pUL21 may have bound to capsids or bound exclusively at the capsid portal vertex that is thought to align the capsid to NPCs. The binding of ectopically expressed pUL16 and pUL21 to incoming capsids is unlikely for at least two reasons. First, the tegument is thought to form through a series of interactions between inner and outer tegument proteins [30] with pUL21 associating with capsids in the nucleus and pUL16 associating with capsids in the cytoplasm after nuclear egress [58, 62–64]. This suggests that the molecular interactions that mediate pUL16 and pUL21 capsid association are distinct, and that pUL16 and pUL21 are not loaded onto capsids as a pUL16/pUL21 complex. Since most of the tegument dissociates from the capsid immediately after entering a cell, it is difficult to conceive that pUL16/pUL21 complexes would bind to incoming capsids that lack molecules involved in pUL16 and pUL21 recruitment. Second, ectopic expression of HSV-2 pUL16 and pUL21 was able to prevent incoming PRV capsid delivery to the nuclear envelope as well as inhibit PRV gene expression (Fig 5). PRV is distantly related to HSV-2, with PRV pUL16 being 28.5% identical to HSV-2 pUL16 and PRV pUL21 being 29.9% identical to HSV-2 pUL21. Thus, the likelihood that HSV-2 pUL16 and pUL21 would be able to interact with incoming PRV capsid components is low.

Since pUL21 interacts with components of the microtubule transport machinery [58, 65], we considered that loss of pUL16 and pUL21 could impact the efficiency with which nascent cytoplasmic nucleocapsids are transported away from the nuclear periphery to sites of secondary envelopment (Fig 10B and 10C). This might enable capsids to linger at the nuclear periphery and provide an enhanced opportunity to engage NPCs. To test this, we prevented nascent cytoplasmic nucleocapsid transport by disrupting microtubules with nocodazole. Nocodazole treatment failed to promote the recruitment of WT capsids to the cytoplasmic face of the nuclear envelope, suggesting that a mechanism(s) remained in place to prevent their recruitment to NPCs (Fig 7C–7E). Importantly, the recruitment of Δ16 and Δ21 capsids to the cytoplasmic face of the nuclear envelope was indistinguishable in the presence or absence of nocodazole, suggesting that the source of capsids recruited to the nuclear envelope was the

infected cell nucleus rather than from superinfecting nucleocapsids that require microtubules for transport to the nucleus.

Analysis of the HSV-1 temperature-sensitive mutant, 50B, demonstrated the accumulation of empty capsids at NPCs late in infection [60]. The authors of this study suggested that the 50B mutant was defective in SIE and that the capsids accumulating at NPCs were derived from superinfecting virions. While our nocodazole experiments (Fig 7) suggested that the source of capsids we observed accumulating at NPCs in cells infected with pUL16 and pUL21 mutants were derived from the infected cell nucleus, rather than superinfecting virions, we nonetheless investigated SIE in this system. No differences in the numbers of capsids entering cells from superinfecting virions were observed between WT, Δ16 and Δ21 infected cells (Fig 4B) suggesting that SIE at the level of virus entry was operating normally in Δ16 and Δ21 infected cells. Previous work examining SIE found that, while their numbers are reduced, capsids from a superinfecting HSV-1 strain are able to enter HSV-1 infected cells; however, an unknown mechanism prevents these cytoplasmic capsids from establishing infection [68]. In agreement with our other findings (Figs 1–3), we found that if cells were initially infected with Δ16 or Δ21, more viral genomes from the superinfecting virus were delivered to the nucleoplasm than in cells initially infected with WT virus (Fig 4D and 4E). It may be that the post-entry SIE observed in previous studies was mediated in whole, or in part, by pUL16 and pUL21 expression during the initial infection.

Finally, we considered that pUL16 and pUL21 may be interacting with NPCs to prevent capsid docking. STED microscopy revealed that both pUL16 and pUL21 localized to the nuclear and cytoplasmic faces of the nuclear envelope in the same plane as lamin A/C, RanBP2 and MAb414 reactive Nups (Figs 8 and 9 and S3). Additionally, pUL21 and pUL16 demonstrated a similar lattice-like distribution in the nuclear envelope and were frequently adjacent to, and colocalized with, NPC components. Thus, if pUL16 and pUL21 interfere with the docking of capsids to NPC components, perhaps by preventing capsid-associated pUL36 and/or pUL25 from binding RanBP2 and Nup214, they are present at the appropriate location within infected and transfected cells to do this (Fig 10B). Alternatively, it may be that pUL16 and pUL21 facilitate the remodeling of NPCs such that they are no longer competent for capsid interactions.

Modifications to NPCs elicited by HSV infection, including dilation of NPCs, clustering of NPCs, alterations in NPC composition and post-translational modification of NPC components, have been reported [69–71]. The dilation of nuclear pores from roughly 100nm in diameter to upwards of 500nm during HSV infection has been suggested to facilitate the egress of nascent HSV nucleocapsids from the nucleoplasm to the cytoplasm [70]. If NPC dilation occurs during HSV infection, then the spatial organization of NPC components could be altered such that nascent cytoplasmic nucleocapsids would be unable to dock at NPCs. However, the NPC dilation hypothesis is difficult to reconcile with the observations that NPC gating function appears unperturbed in HSV infected cells [69]. Moreover, NPC dilation as a means of trafficking capsids from the nucleus to the cytoplasm is incompatible with the observation that DNA-containing capsids, rather than abundant nuclear capsids that lack genomes, are preferentially translocated from the nucleus to the cytoplasm [64, 72]. We did not notice any differences in nuclear pore dimensions between WT, Δ16, Δ21, or Δ21/Δ16 infected cells that could explain the differential recruitment of cytoplasmic nucleocapsids to NPCs (Fig 3D). In addition to reductions in the amounts of RanBP2 associating with other nucleoporins, Hofemeister and O'Hare also noted lower levels of O-glycosylated RanBP2 in HSV-1 infected cells and suggested that alterations in the post-translational modification of pore components during infection might influence nucleocapsid/NPC interactions late in infection [69]. This is an interesting idea that warrants further investigation. It may be that expression of pUL16 and

pUL21 is required to alter RanBP2 glycosylation or influences the ability of RanBP2 to interact with other nucleoporins in a way that is meaningful to capsid/NPC interaction.

In summary, we have provided insight into the mechanism by which HSV averts a short circuit in virion assembly by preventing the attachment of nascent nucleocapsids to NPCs and the delivery of their genomes back into the infected nucleus and, in doing so, have provided answers to a long-standing question in herpesvirus biology.

## Materials and methods

### Cells and viruses

African green monkey kidney cells (Vero), porcine kidney cells (PK15), HeLa cells, life-extended human foreskin fibroblasts (T12) human keratinocytes (HaCaT) and HaCaT cells stably expressing pUL21 or pUL16 (HaCaT21 and HaCaT16, respectively) [38, 44] and life-extended primary human fibroblasts (T12), a gift from Dr. W. A. Bresnahan (University of Minnesota) [73], were maintained in Dulbecco's modified Eagle medium (DMEM) supplemented with 10% fetal bovine serum (FBS) in a 5% $CO_2$ environment. HSV-1 KOS mutants deficient in pUL21 (Δ21), pUL16 (Δ16), or pUL21 and pUL16 (Δ21/Δ16) have been described previously [38, 42, 64]. HSV-2 186 mutants deficient in pUL21 (Δ21), pUL16 (Δ16) have been described previously [34, 43]. Recombinant HSV-2 strain 186 containing mCherry (mCh) fused to the minor capsid protein VP26 and the corresponding Δ21 and Δ16 mutants have been described previously [34, 43]. Recombinant PRV vaccine strain Bartha containing monomeric RFP (mRFP) fused to VP26 (PRV765), a gift from Dr. L. W. Enquist (Princeton University), has been previously described [74] and was propagated in PK15 cells. All pUL21 deficient and pUL16 deficient viruses were propagated in HaCaT21 and HaCaT16 cells, respectively. The HSV-1 KOS Δ21/Δ16 virus was propagated on a 1:1 mixture of HaCaT21 and HaCaT16 cell monolayers as previously described [64]. 5-ethynyl-2'-deoxycytidine (EdC) was incorporated into the genome of HSV-2 186 WT virus as previously described [64] to produce HSV-2 186 WT EdC virus stocks. Times post infection, reported as hours post infection (hpi), refers to the time elapsed following a one-hour inoculation period.

HSV-2 strain 186 carrying pUL21 with mCh fused to its carboxy terminus was constructed by two-step Red-mediated mutagenesis [75], using bacterial artificial chromosome (BAC) clone pYEbac373 [43] in *Escherichia coli* GS1783 [75]. Forward primer 5'-GCTTAC CGTTTGCCTGGCTCGCGCCCAGCACGGCCAGTCTGTGGTGAGCAAGGGCGAG-3' and reverse primer 5'- TGGGTTAGAAAACGACTGCACTTTATTGGGATATCTCACTT GTACAGCTCGTC-3' were used to amplify a PCR product from pEP-mCh-in, a gift from Dr. G.A. Smith (Northwestern University). Restriction fragment length polymorphism analysis following digestion with EcoRI was used to confirm the integrity of BAC DNA. Additionally, PCR was used to amplify products spanning UL21-mCh and these products were sequenced to confirm that mCh was in-frame with UL21 and to ensure that no unintended mutations were introduced. Virus was reconstituted from BAC DNA as described previously [43].

### Plasmids

The following plasmids were used in this study and previously described: enhanced green fluorescent protein (EGFP) pEGFP-N1 (Clontech Laboratories, Mountain View, CA), pCI-neo HSV-2 UL21 (UL21) [43], pCI-neo HSV-2 UL16 (UL16) [34], EGFP fused to the C-terminus of HSV-2 UL21 (UL21-GFP) [43]. To construct a plasmid encoding UL21-mCherry, the EGFP portion of UL21-EGFP was excised by digestion with BamHI and BsrGI and replaced with a BamHI/BsrGI fragment encompassing mCherry excised from pJR70 [76]. To construct a plasmid encoding UL16-EGFP, PCR utilizing forward primer 5'-AGTTC

GAATTCTTATGGCACAGCGGGCACTCTGGCGTCC-3' and reverse primer 5'-GATCGTC GACGGTTTGTAATCGGACGATGAGGCTCTGG-3' were used to amplify the UL16 gene using purified HSV-2 strain HG52 DNA as a template. The product was digested with EcoRI and SalI and ligated into similarly digested pEGFP-N1. To construct a plasmid encoding UL16-mCherry, the UL16 portion of UL16-EGFP was excised by digestion with HindIII and BamHI and ligated to similarly-digested pJR70.

## Immunological reagents and chemicals

Mouse monoclonal antibody against lamin A/C (EMD, Millipore Temecula, CA) was used at a dilution of 1:300, mouse monoclonal antibody against nuclear pore complex proteins (MAb414) (Abcam, Cambridge, MA) was used at a dilution of 1:500, mouse monoclonal antibody against HSV ICP5 (Virusys, Taneytown, MD) was used at a dilution of 1:500, mouse monoclonal antibody against HSV ICP27 (Virusys, Taneytown, MD) was used at a dilution of 1:500, mouse monoclonal antibody against PRV Us3 [77], a gift from Dr. L. W. Enquist (Princeton University) was used at a dilution of 1:500, mouse monoclonal antibody against α-tubulin (GT114) (Invitrogen/Thermo Fisher Scientific, Ottawa, ON) was used at a dilution of 1:500, mouse monoclonal antibody against RanBP2 (D-4) (sc-74518) (Santa Cruz Biotechnology, Dallas, TX) was used at a dilution of 1:50 and rabbit polyclonal antibody against mCherry (Rockland Immunochemicals, Burlington, ON) was used at a dilution of 1:200 for immunofluorescence microscopy. Alexa Fluor 568-conjugated donkey anti-mouse, Alexa Fluor 488-conjugated donkey anti-mouse, Alexa Fluor 647-conjugated donkey anti-mouse were used at a dilution of 1:1000 and Alexa Fluor 488-conjugated donkey anti-rabbit (Thermo Fisher Scientific, Ottawa, ON) was used at a dilution of 1:500, Alexa Fluor 532-conjugated goat anti-mouse (Thermo Fisher Scientific, Ottawa, ON) was used at a dilution of 1:400 and Alexa Fluor 488-conjugated phalloidin (Invitrogen, Ottawa, ON) was used at 165nM for immunofluorescence microscopy. Nocodazole (10mM in DMSO) was used to interfere with cellular microtubule polymerization at a final concentration of 10µM and cells were treated for 6 or 12 hours. AlexaFlour 488 picolyl azide and 5-ethynyl-2'-deoxycytidine (EdC) (Click Chemistry Tools, Scottsdale, AZ) were used for click-chemistry according to the manufacturer's instructions.

## Transmission electron microscopy

Vero cells were seeded onto 100mm dishes one day prior to infection and were infected with HSV-1 KOS WT, Δ21, Δ16 or Δ21/Δ16 virus at a multiplicity of infection (MOI) of 3. At 18 hpi infected cells were processed for transmission electron microscopy as previously described [64]. All micrographs used for data quantification have been uploaded to Dryad and are accessible here [78].

## Immunofluorescence microscopy

Cells that were mock infected or infected with HSV-2 or HSV-1 viruses were fixed at the indicated times post infection by rinsing the cells three times with PBS, followed by fixation in 4% paraformaldehyde in PBS for 15 minutes at room temperature (RT). Cells were then washed three times with PBS/1% BSA and stored in PBS/1% BSA at 4°C until staining. Samples were washed three times in PBS and permeabilized by adding 0.5% Triton X-100 (TX-100) in PBS for 15 minutes at RT. Samples prepared for quantification of nuclei with cytoplasmic capsids surrounding the outer nuclear membrane were permeabilized by adding 0.01% saponin in PBS for 10 minutes at RT. Samples permeabilized with saponin had 0.01% saponin in all subsequent washes and antibody dilutants. After permeabilization, samples were washed three times with PBS/1% BSA and primary antiserum diluted in the appropriate volume of PBS/1% BSA

was applied to samples for one hour at RT. A blocking step was implemented prior to the addition of rabbit primary antibodies using human serum blocking buffer (5% human serum, 0.5% Tween 20 in PBS) for 1 hour at RT. After primary antibody incubation, samples were washed three times with PBS/1% BSA and Alexa Fluor conjugated secondary antibody diluted in the appropriate volume of PBS/1% BSA was applied to samples for 30 minutes at RT. Samples were washed three times with PBS/1% BSA and then incubated with 0.5 μg/mL Hoechst 33342 (Sigma, St. Louis, MO) in PBS for seven minutes at RT. Samples were then washed three times with PBS/1% BSA and stored at 4˚C in PBS/1% BSA until imaging. Samples were imaged through a 60X (1.42 NA) oil immersion objective using an Olympus FV1000 confocal laser scanning microscope and FV10 ASW 4.01 software. Average signal intensity was measured using FV10 ASW 4.01 software.

## Stimulated emission depletion (STED) microscopy

Vero cells were infected with HSV-2 186 UL21-mCh at an MOI of 0.1. At 18 hpi, cells were rinsed three times with PBS followed by fixation in 4% paraformaldehyde in PBS for 15 minutes at RT. Cells were washed three times in PBS and permeabilized in 0.5% Triton X-100 in PBS for 15 minutes at RT. Cells were rinsed three times with PBS/1% BSA and stained for mCh, NPCs (MAb414), or RanBP2, as outlined above. Because mCh was unable to be used STED microscopy, an anti-mCherry antibody was used and detected using an Alexa Fluor 488-conjugated secondary antibody. Confocal and STED microscopy were performed on an SP8 tandem scanning white light laser confocal platform (Leica Microsystems, Richmond Hill, ON) equipped with STED lasers at 592nm and 660nm. A 100X STED objective, NA 1.4, on a DMI 6000 inverted microscope platform with a motorized x, y and z-glavo stage were employed. We used the 590 and 660 STED depletion lasers for Alexa Fluor 488 and Alexa Fluor 532 respectively. All images were deconvolved using Huygens deconvolution software (Hilversum, Netherlands).

## Ectopic expression of pUL16 and pUL21

HeLa or PK15 cells were seeded onto 60mm dishes one day prior to transfection. These cells were then transfected with i) EGFP; ii) EGFP and UL16; iii) EGFP and UL21 or iv) EGFP, UL16 and UL21 expression plasmids using X-treme GENE HP DNA transfection reagent (Roche, Laval, QC) following the manufacturer's instructions. At 6 hours post transfection, samples were incubated in 500μL of complete medium containing 1μL benzonase (250U/μL) for 30min at 37˚C. Cells were then harvested by trypsinization and seeded onto glass coverslips. At 24 hours post transfection, cells were infected on ice for one hour with HSV-2 186 WT mCh-VP26 or PRV765 at an MOI of 3. After one hour, complete medium was added and the cells shifted to 37˚C. At 2 hpi, cells were washed three times in PBS and fixed in 4% paraformaldehyde in PBS for 15 minutes at RT. Samples were then permeabilized with TX-100 and stained with ICP27/Us3 antisera, as outlined in the immunofluorescence microscopy section. The average capsid distance from the nuclear periphery was determined using FV10 ASW 4.01 software. 43–60 capsids per condition were examined.

## Evaluation of pUL21 and pUL16 loading onto incoming capsids

HeLa cells were transfected with EGFP, UL16-EGFP, UL21-EGFP, UL16-EGFP and UL21 or UL21-EGFP and UL16 expression plasmids and prepared for infection as outlined above. At 24 hours post transfection, samples were infected on ice for one hour with HSV-2 186 WT mCh-VP26 at an MOI of 3, shifted to 37˚C following inoculation and fixed at 2 hpi or 18 hpi. Additionally, samples were also infected on ice for one hour with HSV-2 186 Δ16 mCh-VP26

or HSV-2 186 Δ21 mCh-VP26 at an MOI of 3, shifted to 37°C following inoculation and fixed at 18 hpi. Cells were permeabilized and stained for NPCs, as described above. EGFP signal associated with capsids at 2 hpi, as well as the average background EGFP signal (EGFP signal from two points 0.5 μm from the capsid) was measured using FV10 ASW 4.01 software.

### SIE analysis

Vero cells were infected with HSV-2 186 WT, Δ21 or Δ16 virus at an MOI of 1. To examine SIE of superinfecting virus entry and capsid transport to infected nuclei, infected samples were placed on ice for 30 minutes at 6 hpi. At 6.5 hpi, infected samples were superinfected on ice with HSV-2 186 WT mCh-VP26 at an MOI of 3 for 1 hour then shifted to 37°C. At 1 h after the superinfection (8.5 hpi after the initial infection), cells were fixed and stained with Alexa Fluor 488-conjugated phalloidin at a final concentration of 165 nM in PBS for 7 minutes to visualize cortical actin at the cell periphery.

To examine genome delivery of superinfecting virus, infected cells were superinfected at 6 hpi with HSV-2 186 WT EdC labelled virus at an MOI of 3 for 1 hour at 37°C. After inoculation, the inoculum was removed, and cells were incubated in low pH citrate buffer (40mM Na citrate pH 3.68, 10mM KCl, 0.8% NaCl) for 3 minutes at RT to inactivate extracellular virus. Cells were washed three times with complete medium and incubated for 3 h after the superinfection (10 hpi after the initial infection) at which time cells were fixed and EdC detected by click-chemistry using AlexaFlour 488 picolyl azide according to the manufacturer's instructions.

### Statistical analysis

All statistical analyses were performed using GraphPad Prism version 9.1.2.

## Supporting information

**S1 Fig. The colocalization of HSV-2 capsids with NPCs in different cell types.** Cells were infected with HSV-2 186 mCh-VP26 WT, Δ16, or Δ21 virus at an MOI of 0.1 and fixed at 18 hpi. After fixation, cells were permeabilized with TX-100 and stained for NPCs. Representative confocal images of HSV-2 infected T12, HeLa, and HaCaT cells. Merge shows the overlay of Hoechst 33342 (DNA) (blue), NPCs (green), and HSV-2 capsids (red). The scale bar indicates 5μm. Arrowheads indicate cells with mCh-VP26 capsid fluorescence colocalized with NPC fluorescence.
(TIF)

**S2 Fig. SIE control experiments. A)** Representative confocal sections of Vero cells infected on ice with HSV-2 186 mCh-VP26 at an MOI of 3 for one hour and fixed immediately (0 hpi) or after being shifted to 37°C for one hour (1 hpi). Images show Hoechst 33342 (DNA) (blue), phalloidin staining of actin filaments (green) and HSV-2 186 mCh-VP26 capsids (red). The scale bar is 10μm. The arrowhead indicates mCh-VP26 capsids accumulated at the cell surface. **B**) z-projection images of the HSV-2 186 mChVP26 infected cells presented in panel **A**. **C**) Representative image of Vero cells infected on ice with HSV-2 186 WT EdC at an MOI of 3 for one hour and fixed at 0 hpi. Merge shows the overlay of Hoechst 33342 (DNA) (blue) and background Click-chemistry staining (green). The scale bar is 20μm.
(TIF)

**S3 Fig. The localization of pUL16 and pUL21 with respect to lamin A/C in transfected cells.** Cells were imaged by confocal microscopy and stimulated emission depletion (STED) microscopy. Cross section views of the nuclei are show in cells transfected with pUL16-mCh,

pUL16-mCh and pUL21, or pUL21-mCh and pUL16 expression plasmids. Cells were stained with antisera against mCh and lamin A/C. The scale bar is 5μm. Arrowheads indicate cells with pUL16-mCh/pUL21-mCh nuclear rim staining.
(TIF)

## Acknowledgments

We are grateful to Maxwell Sherry for assistance with pUL21-mCh plasmid construction, Dr. Xiaohu Yan, Queen's University, for assistance with TEM, Landon Montag for assistance with statistical analyses, and all members of the Banfield laboratory for helpful comments on the manuscript.

## Author Contributions

**Conceptualization:** Ethan C. M. Thomas, Bruce W. Banfield.

**Data curation:** Bruce W. Banfield.

**Funding acquisition:** Ethan C. M. Thomas, Stephen L. Archer, Bruce W. Banfield.

**Investigation:** Ethan C. M. Thomas, Renée L. Finnen, Jeffrey D. Mewburn, Bruce W. Banfield.

**Methodology:** Ethan C. M. Thomas, Renée L. Finnen, Jeffrey D. Mewburn, Bruce W. Banfield.

**Resources:** Stephen L. Archer, Bruce W. Banfield.

**Supervision:** Bruce W. Banfield.

**Writing – original draft:** Ethan C. M. Thomas, Bruce W. Banfield.

**Writing – review & editing:** Ethan C. M. Thomas, Renée L. Finnen, Jeffrey D. Mewburn, Stephen L. Archer, Bruce W. Banfield.

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
