## [Decision Letter · Decision Letter 0]

18 Aug 2023

Dear Banfield,

Thank you very much for submitting your manuscript "The Herpes Simplex Virus pUL16 and pUL21 Proteins Prevent Capsids from Docking at Nuclear Pore Complexes" for consideration at PLOS Pathogens. As with all papers reviewed by the journal, your manuscript was reviewed by members of the editorial board and by several independent reviewers. In light of the reviews (below this email), we would like to invite the resubmission of a significantly-revised version that takes into account the reviewers' comments.

All three reviewers thought that this is a very exciting study and that the findings are of great importance. Because of the great importance for the field, they felt that the conclusions should be based on solid and well-controlled experimental data. Reviewers #1 and #3 provided a long list of questions and suggestions that will help you improve the manuscript. Most of the questions and concerns can be addressed by changes to the text, re-evaluation (quantification) of existing data, and by making raw image data available to reviewers and readers. However, in a few instances the reviewers ask for additional controls to support the claims (Reviewer #1, points 4 and 9; Reviewer #3, points 8, 10 and 14/15). These issues should be addressed with high priority.

We cannot make any decision about publication until we have seen the revised manuscript and your response to the reviewers' comments. Your revised manuscript is also likely to be sent to reviewers for further evaluation.

Sincerely,

Wolfram Brune

Academic Editor

PLOS Pathogens

Blossom Damania

Section Editor

PLOS Pathogens

Kasturi Haldar

Editor-in-Chief

PLOS Pathogens

orcid.org/0000-0001-5065-158X

Michael Malim

Editor-in-Chief

PLOS Pathogens

orcid.org/0000-0002-7699-2064

All three reviewers thought that this is a very exciting study and that the findings are of great importance. Because of the great importance for the field, they felt that the conclusions should be based on solid and well-controlled experimental data. Reviewers #1 and #3 provided a long list of questions and suggestions that will help you improve the manuscript. Most of the questions and concerns can be addressed by changes to the text, re-evaluation (quantification) of existing data, and by making raw image data available to reviewers and readers. However, in a few instances the reviewers ask for additional controls to support the claims (Reviewer #1, points 3 and 8; Reviewer #3, points 8, 10 and 14/15). These issues should be addressed with high priority.

Reviewer's Responses to Questions

**Part I - Summary**

Reviewer #1: Thomas et al. present evidence for the role of pUL16 and pUL21 in the life cycle of herpes simplex virus 2 (HSV-2) and HSV-1. The manuscript is clearly written and easy to follow. The presented imaging data supports many of the author’s claims, but some need further support. In addition to a general need to make all analyzed data available for review, the authors need to provide further evidence that pUL16 and pUL21 deletion mutants accumulate empty capsids at the cytoplasmic face of NPCs and ideally direct evidence that pUL16 and pUL21 prevent incoming nucleocapsids from docking at NPCs. I encourage publication after a revision as the presented data might explain large parts of how superinfection exclusion works in alphaherpesviruses.

Reviewer #2: In the study entitled 'Prevention of Capsid Docking at Nuclear Pore Complexes by Herpes Simplex Virus pUL16 and pUL21 Proteins', Thomas and colleagues expound upon the mechanisms underlying the interaction between incoming nucleocapsids and Nuclear Pore Complexes (NPCs) during the initial infection phase. They also investigate why nascent nucleocapsids, having undergone nuclear egress, exhibit an inability to dock at NPCs and thereby fail to deliver their genomes to the originating nucleus. The researchers effectively demonstrate that the HSV tegument proteins pUL16 and pUL21, known to form a heterodimeric complex, play a pivotal role in impeding the binding of egressing nucleocapsids to NPCs.

The subject of this manuscript holds considerable importance and garners a wide range of interest. As emphasized by the authors, these findings provide conclusive answers to a longstanding question within herpesvirus biology. Nonetheless, to further fortify the authors' claims, additional experimental evidence is warranted.

Reviewer #3: The authors report an analysis of the potential roles of UL16 and UL21 during HSV infection with results, largely from imaging analysis from which they conclude that a key role of a UL16/UL21 complex is to coat the outer cytoplasmic face of the nuclear membrane, including on nuclear pores, and thus prevent the engagement of progeny HSV capsids with the nuclear pore. Their proposal is that if were this to occur it would in effect would be a self-defeating process for virus replication and release. Thus at least one key role of the UL16/UL21 complex is to prevent this re-engagement and promote forward virus production and release of infectious progeny.

This is a plausible proposition. And if this were indeed the case would prove to be an important new fundamental understanding of the overall processes involved in herpesvirus assembly and release pathway. The strength of the paper is perhaps in the novelty of the proposal which as said would be of fundamental importance.

Nonetheless there are several issues that the authors would need to address both in terms of structure and presentation of the paper and with quantitative analysis of the data themselves

**Part II – Major Issues: Key Experiments Required for Acceptance**

Reviewer #1:

1. all raw image data (image files with microscope metadata) used to quantify phenotypes should be deposited in repositories such as Dryad. Since I do not have access to the data that was used to generate figures 1B-D, 2D, 3B-C, 4B-C,E, 5B-C, E-F, and 6B, I cannot fully evaluate the validity of the claimed findings. See also https://plos.org/open-science/open-data/#accessible-data.

2. some depicted image data is very hard to interpret in the supplied pdf, such as Fig. 4D. The authors should depict single channels in greyscale to make them easier to view. Again, also the underlying raw data for all figures should be made accessible for both the reviewer as well as the reader.

3. it is mostly unclear how image analysis was done for many figures. For example, the authors need to explain what is meant by Fig. 1 in line 875 “cell starting within the nucleus (distance 0) and extending into the cytoplasm”. What point in the nucleus was defined as point 0? How was this made comparable between cells? How were the resulting curves from all cells averaged? Was any alignment done?

4. Fig. 3A/B/C: The provided data indicates that capsids accumulate at the nuclear membrane where NPC is also. At this resolution, both signals often colocalize. However, this does not strictly indicate a direct association. More direct evidence is needed to claim that “ pUL16 and pUL21 prevent incoming nucleocapsids from docking at NPCs,” as stated in the abstract.

5. Fig. 3D: The phenotype in TEM (empty, docked capsids at NPCs) needs to be quantified, and the number of inspected cells needs to be stated if the authors wish to state “that HSV pUL16 and pUL21 deletion mutants accumulate empty capsids at the cytoplasmic face of NPCs late in infection.”(lines 49-50).

6. Fig. 4D: It is unclear how the described experiment was performed. How do the authors exclude capsids that spilled their DNA in the cytoplasm? This signal might be above nuclei but is projected into the same slice. It is unclear which imaging settings were used to exclude this possibility (pinhole size, z-resolution). Also, were whole cells imaged in 3D, or were just single slices taken?

7. Fig. 6 C/D: the “colocalizing signal” indicated to prove the association of UL16 and UL21 with capsids are mostly aggregates and not single particles.

8. Fig. 7: The claimed phenotypes need to be quantified.

9. Fig. 8/9: A control labeling of the inner nuclear membrane is needed if the authors wish to claim that UL21 and UL16 localize to the outer nuclear membrane. The resolution of STED microscopy might not be sufficient alone to determine if a protein is on the inner or outer nuclear membrane. The distance between the two nuclear membranes is 30-50nm. The resolution of a STED microscope typically ranges around 50nm. Antibodies have a size of 10nm and fluorescent proteins 5nm. Summing up these sizes means that the localization precision may not be sufficient for the author’s claims. Immuno-EM might be an alternative approach.

Reviewer #2:

1. In Fig 2A, the differences between WT and Δ16 or Δ21 is not obvious, please state the reason.

2. The authors should perform some experiments in the epithelial cells, such as HFF or HEL.

Reviewer #3:

1. The introduction to this work is a quite simplistic overview of HSV replication and exit pathways and reads a little bit like a student project summary. Considering this is about the function of UL16 and UL21, the introduction could do with a much better detailed and relevant introduction to these proteins and any previous data on their roles and importance for overall replication. This links to point 2.

2. The first phenotype that we are presented with for the deletion mutants for UL16 and UL21 is on the localization of the major capsid protein VP5. There is no data on the effect of these deletions on overall replication of the virus and if this has not previously been published upon by the authors then clearly this would be amongst the first most relevant data to present. Conversely if they have reported previously on the effect of these deletions on overall replication then that obviously should be introduced. But perhaps more relevant would be the preliminary discussion of the effect of deletion of UL16 and or UL21 not only on overall replication of HSV, but on the effect on extracellular versus cytoplasmic infectious virus. There could be an effect on both, there could be an effect only on cytoplasmic infectious virus. There could even be a result that, despite the validity of preventing re-engagement of progeny virus with the nuclear pore, this actually made a minor impact on the overall abundance of released virus. Alternatively, if the deletion mutants each essentially abolished replication having a profound impact on the production of infectious virus, either way the relevant data should be either introduced and or presented early.

3. The first data we're given is on the distribution of a fluorescent fusion protein mCh-VP26, (a structural capsid protein), in cells infected with wild type virus versus the deletion mutants. The key phenotype here is on the distribution of mChVP26 to the outside nuclear rim. In the text, (page 5, line 132/133), this is described as ..” An intense ring of mChVP-26 Fluorescence at the nuclear periphery.” While there does appear to be a phenotype with increased nuclear rim localization in this and other figures, this localization can appear rather patchy and certainly not in an intact intense ring. For example, even in the first figure for the UL21 deletion mutant the fluorescent fusion protein is mainly located in very large aggregates inside the nucleus, which themselves seem rather variable throughout the paper, with a minor population at the nuclear rim, and in patches. The same is true for the mutants in figure 2A and the HSV 1 versions in figure 2B and elsewhere. This may well be the proposed phenotype but perhaps it should be more accurately described, and this leads to another point on the way it is classified and quantified.

4. The authors state, page 7, line 166, “The percentage of sales with an abundance of capsids at the cytoplasmic face of the nuclear envelope was quantified”. This is then given as a percentage of nuclei, i.e., a single nucleus will be scored positive or not positive. But what exactly this means is not clear. For example, in many of the images it is not the case that majority of the capsid signal is at the nuclear envelope, in many cases it is clearly within the nucleus and in several in large nuclear aggregates. This may ultimately be an understandably difficult classification, but some more accurate description of the classification could be given.

5. In this vein the authors have an opportunity to strengthen the interpretation of the phenotype from their transmission electron microscopy in figure 3. They suggest, page 8, line 187, that for the wild type virus at 18 hours after infection it was “very rare” to see capsids docked at the nuclear pore, while this was “common” for the UL16 and UL21 mutant viruses. This could also be quantified, and indeed normalised; for example number of NPC docked capsids/total cytoplasmic capsids, for each of the wild type and mutant viruses. This would then support and supplement the classification from the immunofluorescence studies.

6. In Figure 4 the authors then use the idea of superinfection exclusion to support their proposition. The idea is that prior infection with wild type HSV would, by virtue of its expression of UL16 and UL21, block the nuclear docking and genome entry of a subsequent super-infecting virus. On a minor point the description of this experiment in the legend to Figure 4 seems at odds with the description given in materials and methods. In the legend the Superinfecting virus is stated as being at an MOI of 1, while stated as an MOI of three in the materials and methods. In the legend the analysis of the Superinfecting virus is at one hour after infection, while in the materials and methods, superinfection is for one hour on ice followed by a shift to 37 degrees and subsequent analysis two hours later. While this might seem minor the detail is important for the following points.

7. Figure 4a includes a normal infection of wild-type into previously uninfected cells (mock). The numbers of capsids on the cell appeared to be in the hundreds, and even with a relatively poor particle infectivity, this seems a very high particle count.

8. In this regard these are maximum projections and thus very likely will measure viruses/capsids bound to the outside of cells (indeed that is why there is such a very high particle count). Given that, following the Superinfection and shift to 37 degrees for two hours, the authors propose their examining internalised capsids. How do they know? There are no controls. Minimally the authors should superinfect for the one hour on ice, and fix and sample there, without any shift up in temperature. It may well be that a quite a high proportion of what they measure is at the cell surface reflecting virus that has not or will not enter cells.

9. Having said that the authors are not claiming any significant phenotype of primary infection by the mutant viruses in affecting the cytoplasmic entry or localization to the nuclear periphery of the Superinfecting virus.

10. What they do claim is that there is a phenotype in allowing nuclear pore docking and genome entry into the nucleus of the Superinfecting virus. The idea is that a primary infection by a wild-type virus suppresses genome entry of the Superinfecting virus, while a primary infection by either of the UL16 or UL21 mutant viruses has much less of an effect in this regard, thus permitting significantly more genome entry of the Superinfecting virus. However the imaging data are not particularly clear and no controls are presented. The images are proposed to represent uncoated EdC-labelled genomes entering the nucleus after super infection. Quantitation of this indicates that under conditions where the primary infection is by wild-type virus, for the super-infecting virus there would be approximately one genome observed for every two cells on average. When the primary infection is by the UL16 or UL21 deletion mutants, then for the super-infecting virus there is approximately 1 genome for every one cell (or very slightly more). Notwithstanding the numbers of cells stated for the quantification, this seems a very marginal effect that is compounded by difficulty interpreting the images themselves. While there are some new clear foci that are proposed to represent uncoated nuclear genomes, there is also quite a lot of both diffuse membrane-like cytoplasmic and diffuse nucleoplasmic signal including what looks to be nucleolar signals in some cells. It would be useful to have some controls, even if presented a supplementary information, for example examining cells by the click chemistry reaction but where infection was only at 4 degrees, thus preventing entry and nuclear uncoating. Perhaps the background would remain but the nuclear foci it would disappear confirming their interpretation. Another piece of supplementary information that might influence the design of the assay would be showing increasing numbers of nuclear foci with multiplicity of infection of the super-infecting virus. However it might be easier to demonstrate the point with a different assay. The quantitation indicates that, Where the primary infection is wild type, only one in two cells would be superinfected. While where the primary infection is a mutant virus, all cells will be super infected. perhaps an easier and justice robust assay would be measuring very early protein expression, for example ICP4 or ICP0, if necessary using a fluorescent- fusion protein to discriminate the superinfecting virus. Another example of an experiment that would help support the proposal, is it the abundance of expression of UL16 and UL21 from the primary infection would correlate The decrease in the efficiency of super-infection. it might be possible to examine this, either by the nuclear genome entry assay or by the superinfecting protein expression assay.

11. In figure 5, pursuing the idea that prior expression of UL16 or UL21 would interfere with super infection, the authors ask topically express these proteins and then measure the influence on capsid recruitment to the proximity of the nuclear envelope. While in figure 4A the similar experiment shows extremely high numbers of capsids at various distances and locations in the infected cell, it is not clear why in the similar experiment in figure 5, there appeared to be very low numbers come on a few at most during infection of in this case the previously transfected cells. Nonetheless the experiments should have a very similar framework.

12. The paragraph, page 13 line 292, is a little confusing. it starts by confirming that UL16 and UL21 can interact with capsids, and finishes with the conclusion that UL16 and UL21 do not interact with incoming capsids, and thus are unlikely to prevent capsid recruitment to the nuclear pore. This needs a little better clarification and perhaps at this point more explicit proposal of what they are likely to do in preventing superinfection.

13. The experiment shown in figure 7 or actually some of the better images presented showing the difference in capsid protein recruitment to the nuclear rim by the mutants versus very little observed for the wild type virus.

14. In Figure 8 the authors use STED for more refined imaging analysis of the distribution of UL21 with regards to the nuclear envelope and in particular whether there is a significant relative accumulation on the cytoplasmic face. There are particularly convincing data here especially for the Cross sections with the MAB414. it would be useful to know how it was defined that they were examining the cytoplasmic face, and might not a Z- projection be useful in this regard. an obvious question also is whether they or indeed anyone else has reported on the nuclear rim localization of UL21 using antibody to endogenous protein.

15. With regards to Figure 9 the proposal for some population of UL16-cCh residing specifically at the outer nuclear rim, would be enhanced by the appropriate control of UL16-mCh alone, especially by STED, from where they are proposing a lattice like appearance. It is odd that the internal UL16 and indeed UL21 do not appear to be visualised in the cross section. is there an explanation for this and in particular when they do visualise the nuclear internal proteins do they also appear as a lattice like pattern in the STED. If there is a clear qualitative difference this would reinforce their case.

**Part III – Minor Issues: Editorial and Data Presentation Modifications**

Reviewer #1: -Fig. 4B would be easier to read if expressed in percent of mock.

-The number of analyzed cells for Fig. 3B and C is missing.

- line 515-516: The ΔpUL16/pUL21 virus was grown on a 1:1 mixture of pUL21 and pUL16-complementing cells. These conditions could lead to a mixed population of viruses in the stock produced.

- In lines 570-572: Duration of treatment of Nocodazole missing in the methods.

- In lines 620—622, it is hard to follow which plasmids were transfected together.

Reviewer #2: No

Reviewer #3: point 1 above

PLOS authors have the option to publish the peer review history of their article (what does this mean?). If published, this will include your full peer review and any attached files.

Reviewer #1: No

Reviewer #2: No

Reviewer #3: No

Figure Files:

Data Requirements:

Please note that, as a condition of publication, PLOS' data policy requires that you make available all data used to draw the conclusions outlined in your manuscript. Data must be deposited in an appropriate repository, included within the body of the manuscript, or uploaded as supporting information. This includes all numerical values that were used to generate graphs, histograms etc.. For an example see here on PLOS Biology: http://www.plosbiology.org/article/info:doi%2F10.1371%2Fjournal.pbio.1001908#s5.
---

## [Decision Letter · Decision Letter 1]

31 Oct 2023

Dear Banfield,

Thank you very much for submitting your manuscript "The Herpes Simplex Virus pUL16 and pUL21 Proteins Prevent Capsids from Docking at Nuclear Pore Complexes" for consideration at PLOS Pathogens. As with all papers reviewed by the journal, your manuscript was reviewed by members of the editorial board and by several independent reviewers. The reviewers appreciated the attention to an important topic. Based on the reviews, we are likely to accept this manuscript for publication, providing that you modify the manuscript according to the review recommendations.

While you have addressed the comments of reviewers #1 and #2 adequately, reviewer #3 remains unsatisfied. Reviewer #3 points out a number of remaining issues, which can all be addressed without additional experiments. If decide to reject specific reviewer requests, please explain in your point-by-point response why you disagree.

Sincerely,

Wolfram Brune

Academic Editor

PLOS Pathogens

Blossom Damania

Section Editor

PLOS Pathogens

Kasturi Haldar

Editor-in-Chief

PLOS Pathogens

orcid.org/0000-0001-5065-158X

Michael Malim

Editor-in-Chief

PLOS Pathogens

orcid.org/0000-0002-7699-2064

While you have addressed the comments of reviewers #1 and #2 adequately, reviewer #3 remains unsatisfied. Reviewer #3 points out a number of remaining issues, which can all be addressed without additional experiments. If decide to reject specific reviewer requests, please explain in your point-by-point response why you disagree.

Reviewer Comments (if any, and for reference):

Reviewer's Responses to Questions

**Part I - Summary**

Reviewer #1: Thomas et al. provide a thorough revision of their manuscript. I appreciate added data as well as uploading the raw data into a public repository. I have no further concerns regarding the manuscript and suggest publication.

Reviewer #3: It is an interesting and unexplored proposal that there may be a specific mechanism that limits the recruitment of progeny herpesvirus capsids back to the nuclear pores of the infected cells in which they are produced. This work concerns such a proposal in herpes simplex virus and results are presented that the HSV proteins UL21 and UL16 for a complex that localises to the nuclear envelope/pores and somehow blocks capsid recruitment/genome uncoating at the pores. Some sound data is presented in support of this proposition but more explanation on precisely how the data are quantified and on presentation to help precision and help support the conclusions

**Part II – Major Issues: Key Experiments Required for Acceptance**

Reviewer #1: (No Response)

Reviewer #3: The authors have gone some way to addressing the key points raised in response to the original manuscript. However certain points have not been adequately addressed or require further clarification.

1. With regards to original points 1 and 2, on providing some more and relevant introductory detail: While it might be case that a generalised background helps context, what would certainly help context, relevant to the details of this paper, would be more introductory information specifically on UL16 and UL21. Also stated, if the authors are interested in context, it would be extremely useful to mention in more detail on the extent of the impact of UL16 and UL21 deletion mutants, on replication, especially on intracellular versus extracellular yield. Their response to this point it really pays it lip service and doesn't serve the reader well when they state o the mutants, line 115, “to varying degrees, all of these strains suffer defects in virus replication”. Although they also point to the materials and methods indicating that some detail had been given there, in fact this simply refers back to the previous papers again. Referring back to previous papers in this way without more information as requested really doesn't aid context for the reader. They should give a succinct and clear statement of the extent of the phenotype on replication of intracellular and extracellular virus of these mutants, including differences in eg HSV-1 and HSV-2, to provide the context they apparently wish to provide.

2. The original point, (4), referred to figure 2C and its quantitation in figure 2D. This latter figure assesses VP5 localisation as a proxy for capsids, and is plotted as ” nuclei with capsids surrounding the nuclear envelope”. This was given as a percentage of total nuclei. The query was on clarification of exactly what was being measured and how. The authors have now stated that this measurement solely concerns the analysis performed with saponin permeabilization for imaging, i.e. where there is no background VP5 detected in the nucleus. However, some clarification is still required. For example, in figure 2C, zoomed panels are illustrated presumably to help explain what the authors are scoring as positive and negative. With regard to the measurement, “nuclei with capsids surrounding the nuclear envelope”, they score the single image in figure 2C as negative for the wild type, but the image for the deletion mutants as positive. At least to this reviewer I cannot see the difference. The arrowheads added to the panels for the mutants aren't really explanatory and it is unclear why this is classed as different from the wild type image. More clarification is still required.

3. A related point relates to the quantitation of these images, -and the proposal that the lack of UL16 or UL21 allows this capsid recruitment to the nuclear pore/periphery. From the quantitation this phenotype appears to be minor. While it they state that the assay has essentially no background, in other words, no wild type infected cells appear to show cytoplasmic VP5 in proximity to the nuclear envelope, nevertheless from the data, only approximately 1% of the single mutant viruses show this phenotype and are scored positive. Thus 99% of the mutant infected cells do not show capsid accumulation at the nuclear periphery. If this is the correct interpretation of their data, the authors should comment on this. Alternatively, if this is not the interpretation the authors wish to leave, they should clarify.

4. With regards to figure 5D, the authors indicate that prior ectopic expression of UL16 and UL21, results in a decreased expression of ICP27 from an incoming virus, despite there being detectable incoming capsids in the cytoplasm. This is apparently shown on the inset for figure 5D. However, it is not readily apparent on the figure what they are referring to, nor is there anything readily visible, or pointed to with arrows, in the corresponding mCH-VP26 channel. The authors should clarify and label very precisely with appropriate arrowheads.

5. In figure 6 the authors indicate that while independent expression of UL16 shows a broad heterogeneous pattern of localization, co-expression with UL21, results in, “pUL16-GFP also concentrated at the nuclear envelope”. This, as with several other statements in the results, is an overemphasis: rather that a minor population of overall UL16 appeared to localise to the nuclear rim when co-expressed with UL21. The authors should correct the description that UL16 concentrated at the nuclear envelope.

6. In the same experiment the authors indicate that transfected and ectopically expressed UL16-GFP is recruited to and co-localises with cytoplasmic capsids produced later after w/t infection of the transfected cells Fig 6c, w/t). They indicate this with an arrow on the corresponding GFP panel. it is however extremely difficult to determine what is the capsid signal they are indicating that colocalises with this arrowed UL16-GFP region. It is not indicated on the mCH-VP26 panel, and I can determine very little mCH-VP26 signal there. In fact there doesn't seem to be much of a cytoplasmic capsid signal at all, rather the mCH-VP26 fusion protein appears to be in large nuclear aggregates. And there is certainly a lot of the UL16-GFP green signal without any VP26 association. The authors should clarify the figure and indicate the co-localising capsids much more precisely and clearly.

7. Line 306, clarification needed. After the results indicating that ectopically expressed pUL16 and pUL21 can be recruited onto late assembling capsids the authors state in the same paragraph, “Collectively, these data suggest that , when ectopically expressed, pUL16 and pUL21 are not loaded onto incoming capsids “. But this mainly refers back to the previous paragraph and Figure 6a/a. For clarity the authors should state “Collectively, the data of Figure 6, suggest that , when ectopically expressed, pUL16 and pUL21 are not loaded onto incoming capsids…….”

8. (Original point 14). With reference to pUL21 localisation in Figure 8, the query was whether other pertinent previous data on UL21 were available. The authors respond that there are, but they don’t include them at the appropriate juncture, nor indeed cite the relevant information from e.g. Benedyk et al. For example it would improve this paper if the authors acknowledged both their own and others work relevant to figure 8 by initially stating something like, line 339, “previous work from our own and other laboratories has demonstrated that a population of UL21 exhibits a nuclear rim like localization (refs). We next wished to pursue a more refined analysis of both UL21 and UL16 localization and in particular more precise localization with regards to the cytoplasmic and or nuclear face of the nuclear envelope” or something similar.

9. Finally, coming back to the first point on putting this study in context, the authors state that their very basic description of the HSV life cycle, is to put the study in context for the reader. I suggest that readers of the manuscript will likely have a reasonable knowledge to the herpesvirus life cycle and that the reader would be better served by giving context regarding previous data not only on the extent or otherwise of any deficiency in actual virus replication (eg no defect for HSV-1, versus essential for HSV-2), any inconsistency in those phenotypes, but also information from previous publications from others (and indeed their own) on possible roles of these proteins. For example, pUL21 is reported as being required for efficient capsid nuclear egress, secondary envelopment and virus spread. But the requirement for pUL16 and pUL21 in nuclear egress varies depending on strain and cell type. In addition to promoting capsid translocation, the pUL21:pUL16 complex may promote wrapping of cytoplasmic virions via interactions with other viral proteins. pUL21 may also promote virus spread, via an interaction with gE. pUL21 has also been observed to promote the formation of long cellular processes, potentially by regulating microtubule polymerisation. It has also been reported to be an adaptor of protein phosphatase an affects phosphorylation of both cellular and viral nuclear egress proteins. Perhaps not all of this is related to this present study, but certainly some attempt should be made, instead of a rather basic introduction, to give context with information both on the effect of deletion of these proteins on virus yield (but more than a cursory back reference without stating the actual result) and prior art of multiplicity of functions of these proteins.

**Part III – Minor Issues: Editorial and Data Presentation Modifications**

Reviewer #1: (No Response)

Reviewer #3: see above

PLOS authors have the option to publish the peer review history of their article (what does this mean?). If published, this will include your full peer review and any attached files.

Reviewer #1: No

Reviewer #3: No

Figure Files:

Data Requirements:

Please note that, as a condition of publication, PLOS' data policy requires that you make available all data used to draw the conclusions outlined in your manuscript. Data must be deposited in an appropriate repository, included within the body of the manuscript, or uploaded as supporting information. This includes all numerical values that were used to generate graphs, histograms etc.. For an example see here: http://www.plosbiology.org/article/info:doi%2F10.1371%2Fjournal.pbio.1001908#s5.

Reproducibility:

References:

---

## [Editor Report · Decision Letter 2]

16 Nov 2023

Dear Banfield,

We are pleased to inform you that your manuscript 'The Herpes Simplex Virus pUL16 and pUL21 Proteins Prevent Capsids from Docking at Nuclear Pore Complexes' has been provisionally accepted for publication in PLOS Pathogens.

Best regards,

Wolfram Brune

Academic Editor

PLOS Pathogens

Blossom Damania

Section Editor

PLOS Pathogens

Kasturi Haldar

Editor-in-Chief

PLOS Pathogens

orcid.org/0000-0001-5065-158X

Michael Malim

Editor-in-Chief

PLOS Pathogens

orcid.org/0000-0002-7699-2064
---

## [Editor Report · Acceptance letter]

27 Nov 2023

Dear Banfield,

We are delighted to inform you that your manuscript, "The Herpes Simplex Virus pUL16 and pUL21 Proteins Prevent Capsids from Docking at Nuclear Pore Complexes," has been formally accepted for publication in PLOS Pathogens.

Best regards,

Kasturi Haldar

Editor-in-Chief

PLOS Pathogens

orcid.org/0000-0001-5065-158X

Michael Malim

Editor-in-Chief

PLOS Pathogens

orcid.org/0000-0002-7699-2064